# UNCERTAINTY-AWARE PREDICTION FOR GRAPH NEURAL NETWORKS

## ABSTRACT

Thanks to graph neural networks (GNNs), semi-supervised node classification has shown the state-of-the-art performance in graph data. However, GNNs have not considered different types of uncertainties associated with the class probabilities to minimize risk increasing misclassification under uncertainty in real life. In this work, we propose a Subjective Bayesian deep learning framework reflecting various types of uncertainties for classification predictions by leveraging the powerful modeling and learning capabilities of GNNs. We considered multiple uncertainty types in both deep learning (DL) and belief/evidence theory domains. We treat the predictions of a Subjective Bayesian GNN (S-BGNN) as nodes' multinomial subjective opinions in a graph based on Dirichlet distributions where each belief mass is a belief probability of each class. By collecting evidence from the given labels of training nodes, the S-BGNN model is designed for accurately predicting probabilities of each class and detecting out-of-distribution. We validated the outperformance of the proposed S-BGNN, compared to the state-of-the-art counterparts in terms of the accuracy of node classification prediction and out-of-distribution detection based on six real network datasets.

## 1 INTRODUCTION

Inherent uncertainties introduced by different root causes have emerged as serious hurdles to find effective solutions for real world problems. Critical safety concerns have been brought due to lack of considering diverse causes of uncertainties, resulting in high risk due to misinterpretation of uncertainties (e.g., misdetection or misclassification of an object by an autonomous vehicle). Graph neural networks (GNNs) (Kipf & Welling, 2016; Veličković et al., 2018) have gained tremendous attention in the data science community. Despite their superior performance in semi-supervised node classification and/or regression, they didn't allow to deal with various types of uncertainties. Predictive uncertainty estimation (Malinin & Gales, 2018) using Bayesian NNs (BNNs) has been explored for classification prediction or regression in the computer vision applications, with well-known uncertainties, aleatoric and epistemic uncertainties. Aleatoric uncertainty only considers data uncertainty derived from statistical randomness (e.g., inherent noises in observations) while epistemic uncertainty indicates model uncertainty due to limited knowledge or ignorance in collected data. On the other hand, in the belief or evidence theory, Subjective Logic (SL) (Josang et al., 2018) considered vacuity (or lack of evidence) as uncertainty in an subjective opinion. Recently other uncertainties such as dissonance, consonance, vagueness, and monosonance (Josang et al., 2018) are also introduced.

This work is the first that considers multidimensional uncertainty types in both DL and belief theory domains to predict node classification and out-of-distribution (OOD) detection. To this end, we incorporate the multidimensional uncertainty, including vacuity, dissonance, aleatoric uncertainty, and epistemic uncertainty in selecting test nodes for Bayesian DL in GNNs. We perform semi-supervised node classification and OOD detection based on GNNs. By leveraging the modeling and learning capability of GNNs and considering multidimensional uncertainties in SL, we propose a Bayesian DL framework that allows simultaneous estimation of different uncertainty types associated with the predicted class probabilities of the test nodes generated by GNNs. We treat the predictions of a Subjective Bayesian GNN (S-BGNN) as nodes' subjective opinions in a graph modeled as Dirichlet distributions on the class probabilities, and learn the S-BGNN model by collecting the evidence from the given labels of the training nodes (see Figure 1). This work has the following **key contributions**:

- **A Subjective Bayesian framework to predictive uncertainty estimation for GNNs**. Our proposed framework directly predicts subjective multinomial opinions of the test nodes in a graph,

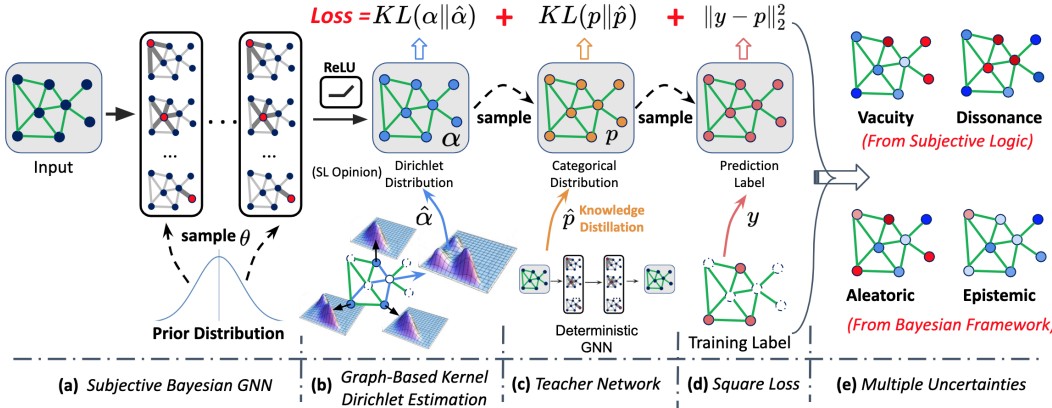

Figure 1: **Method Overview.** our proposed framework is based on (a) Subjective Bayesian GNN designed for estimating the different types of uncertainties including (e) vacuity, dissonance, aleatoric, and epistemic uncertainties for the applications in graph data. The loss function includes (d) square error to reduce bias and (b) (c) two KL components to reduce error in predicting uncertainty.

with the opinions following Dirichlet distributions with each belief probability as a class probability. Our proposed framework is a generative model, so it cal be highly applicable across all GNNs and allows simultaneously estimating different types of associated uncertainties with the class probabilities.

- **Efficient approximate inference algorithms**: We propose a Graph-based Kernel Dirichlet distribution Estimation (GKDE) method to reduce error in predicting Dirichlet distribution. We designed an iterative knowledge distillation algorithm that treats a deterministic GNN as a teacher network while considering our proposed Subjective Bayesian GNN model (a realization of our proposed framework for a specific GNN) as a distilled network. This allows the expected class probabilities based on the predicted Dirichlet distributions (i.e., outputs of our trained Bayesian model) to match the predicted class probabilities of the deterministic GNN model, along with uncertainty estimated in the predictions.

- **Comprehensive experiments for the validation of the performance of our proposed framework**. Based on six real graph datasets, we compared the performance of our propose framework with that of other competitive DL algorithms. For a fair comparison, we tweaked the DL algorithms to consider various uncertainty types in predicted decisions.

## 2  RELATED WORK

**Epistemic Uncertainty in Bayesian Deep Learning (BDL)**: Machine/deep learning (M/DL) research mainly considered *aleatoric* uncertainty (AU) and *epistemic* uncertainty (EU) using BNNs for computer vision applications. AU consists of homoscedastic uncertainty (i.e., constant errors for different inputs) and heteroscedastic uncertainty (i.e., different errors for different inputs) (Gal, 2016). A BDL framework was presented to estimate both AU and DU simultaneously in regression settings (e.g., depth regression) and classification settings (e.g., semantic segmentation) (Kendall & Gal, 2017). Later, a new type of uncertainty, called *distributional uncertainty* (DU), is defined based on distributional mismatch between the test and training data distributions (Malinin & Gales, 2018). *Dropout variational inference* (Gal & Ghahramani, 2016) is used as one of key approximate inference techniques in BNNs. Other methods (Eswaran et al., 2017; Zhang et al., 2018) measure overall uncertainty in node classification but didn't consider uncertainty decomposition and GNNs.

**Uncertainty Quantification in Belief/Evidence Theory**: In the belief/evidence theory domain, uncertainty reasoning has been substantially explored, such as Fuzzy Logic (De Silva, 2018), Dempster-Shafer Theory (DST) (Sentz et al., 2002), or Subjective Logic (SL) (Jøsang, 2016). Belief theory focuses on reasoning of inherent uncertainty in information resulting from unreliable, incomplete, deceptive, and/or conflicting evidence. SL considered uncertainty in subjective opinions in terms of *vacuity* (i.e., lack of evidence) and *vagueness* (i.e., failing in discriminating a belief state) (Jøsang, 2016). Recently, other uncertainty types have been studied, such as *dissonance* (due to conflicting evidence) and *consonance* (due to evidence supporting composite states) (Josang et al., 2018).

In deep NNs, SL is considered to train a deterministic NN for supervised classification in computer vision applications (Sensoy et al., 2018). However, they didn't consider a generic way of estimating multidimensional uncertainty using Bayesian DL for GNNs used for the applications in graph data.

## 3 PROPOSED APPROACH

Now we define the problem of *uncertainty-aware semi-supervised node classification* and then present a Bayesian GNN framework to address the problem.

### 3.1 PROBLEM DEFINITION

Given an input graph $\mathcal{G} = (\mathbb{V}, \mathbb{E}, \mathbf{r}, \mathbf{y}_{\mathbb{L}})$, where $\mathbb{V} = \{1, \cdots, N\}$ is a ground set of nodes, $\mathbb{E} \subseteq \mathbb{V} \times \mathbb{V}$ is a ground set of edges, $\mathbf{r} = [\mathbf{r}_1, \cdots, \mathbf{r}_N]^T \in \mathbb{R}^{N \times d}$ is a node-level feature matrix, $\mathbf{r}_i \in \mathbb{R}^d$ is the feature vector of node $i$, $\mathbf{y}_{\mathbb{L}} = \{y_i \mid i \in \mathbb{L}\}$ are the labels of the training nodes $\mathbb{L} \subset \mathbb{V}$, and $y_i \in \{1, \ldots, K\}$ is the class label of node $i$. **We aim to predict**: (1) the **class probabilities** of the testing nodes: $\mathbf{p}_{\mathbb{V} \setminus \mathbb{L}} = \{\mathbf{p}_i \in [0, 1]^K \mid i \in \mathbb{V} \setminus \mathbb{L}\}$; and (2) the **associated multidimensional uncertainty estimates** introduced by different root causes: $\mathbf{u}_{\mathbb{V} \setminus \mathbb{L}} = \{\mathbf{u}_i \in [0, 1]^m \mid i \in \mathbb{V} \setminus \mathbb{L}\}$, where $p_{i,k}$ is the probability that the class label $y_i = k$ and $m$ is the total number of uncertainty types.

### 3.2 MULTIDIMENSIONAL UNCERTAINTY QUANTIFICATION

Multiple uncertainty types may be estimated, such as *aleatoric uncertainty*, *epistemic uncertainty*, *vacuity*, *dissonance*, among others. The estimation of the first two types of uncertainty relies on the design of an appropriate Bayesian DL model with parameters, $\boldsymbol{\theta}$. Following (Gal, 2016), node $i$'s **aleatoric uncertainty** is: $\text{Aleatoric}[\mathbf{p}_i] = \mathbb{E}_{\text{Prob}(\boldsymbol{\theta}|\mathcal{G})}\big[\mathcal{H}(\mathbf{y}_i|\mathbf{r}; \boldsymbol{\theta})\big]$, where $\mathcal{H}(\cdot)$ is Shannon's entropy of $\text{Prob}(\mathbf{p}_i|\mathbf{r}; \boldsymbol{\theta})$. The **epistemic uncertainty** of node $i$ is estimated by:

$$\text{Epistemic}[\mathbf{p}_i] = \mathcal{H}\big[\mathbb{E}_{\text{Prob}(\boldsymbol{\theta}|\mathcal{G})}[(\mathbf{y}_i|\mathbf{r}; \boldsymbol{\theta})]\big] - \mathbb{E}_{\text{Prob}(\boldsymbol{\theta}|\mathcal{G})}\big[\mathcal{H}(\mathbf{y}_i|\mathbf{r}; \boldsymbol{\theta})\big] \tag{1}$$

where the first term indicates **entropy** (or **total uncertainty**).

*Vacuity* and *dissonance* can be estimated based on the subjective opinion for each testing node $i$ (Josang et al., 2018). Denote $i$'s subjective opinion as $[b_{i1}, \cdots, b_{iK}, v_i]$, where $b_{ik}(\geq 0)$ is the belief mass of the $k$-th category, $v_i(\geq 0)$ is the uncertainty mass (i.e., **vacuity**), and $K$ is the total number of categories, where $\sum_{k=1}^{K} b_{ik} + v_i = 1$. $i$'s **dissonance** is obtained by:

$$\omega(b_i) = \sum_{k=1}^{K} \Big( \frac{b_{ik} \sum_{j=1, j \neq k}^{K} b_{ij} \text{Bal}(b_{ij}, b_{ik})}{\sum_{j=1, j \neq k}^{K} b_{ij}} \Big), \tag{2}$$

where the relative mass balance between a pair of belief masses $b_{ij}$ and $b_{ik}$ is expressed by $\text{Bal}(b_{ij}, b_{ik}) = 1 - |b_{ij} - b_{ik}|/(b_{ij} + b_{ik})$. To develop a Bayesian GNNs framework that predicts multiple types of uncertainty, we estimate *vacuity* and *dissonance* using a Bayesian model. In SL, a multinomial opinion follows a Dirichlet distribution, $\text{Dir}(\mathbf{p}_i|\boldsymbol{\alpha}_i)$, where $\boldsymbol{\alpha}_i \in [1, \infty]^K$ represents the distribution parameters. Given $S_i = \sum_{k=1}^{K} \alpha_{ik}$, belief mass $\mathbf{b}_i$ and uncertainty mass $v_i$ can be obtained by $b_{ik} = (\alpha_{ik} - 1)/S_i$ and $v_i = K/S_i$.

### 3.3 PROPOSED BAYESIAN DEEP LEARNING FRAMEWORK

Let $\mathbf{p} = [\mathbf{p}_1, \ldots, \mathbf{p}_N]^\top \in \mathbb{R}^{N \times K}$ denote the class probabilities of the node in $\mathbb{V}$, where $\mathbf{p}_i = [p_{i1}, \ldots, p_{iK}]^\top$ refers to the class probabilities of a specific node $i$. As shown in Figure 1, our proposed Bayesian GNN framework can be described by the generative process:

- Sample $\boldsymbol{\theta}$ from a predefined prior distribution, i.e., $\mathcal{N}(\mathbf{0}, \mathbf{I})$.
- For each node $i \in \mathbb{V}$: (1) Sample the class probabilities $\mathbf{p}_i$ from a Dirichlet distribution: $\text{Dir}(\mathbf{p}_i|\boldsymbol{\alpha}_i)$, where $\boldsymbol{\alpha}_i = f_i(\mathbf{r}; \boldsymbol{\theta})$ is parameterized by a GNN network $\boldsymbol{\alpha} = f(\mathbf{r}; \boldsymbol{\theta}) : \mathbb{R}^{N \times d} \to [1, \infty]^{N \times K}$ that takes the attribute matrix $\mathbf{r}$ as input and directly outputs all the node-level Dirichlet parameters $\boldsymbol{\alpha} = [\boldsymbol{\alpha}_1, \cdots, \boldsymbol{\alpha}_N]$, and $\boldsymbol{\theta}$ refer to the hyper-parameters of the GNN network; and (2) Sample $y_i \sim \text{Cat}(y_i|\mathbf{p}_i)$, a categorical distribution on $\mathbf{p}_i$.

In this design, the graph dependencies among the class labels in $\mathbf{y}_{\mathbb{L}}$ and $\mathbf{y}_{\mathbb{V} \setminus \mathbb{L}}$ are modeled via the GNN network $f(\mathbf{r}; \boldsymbol{\theta})$. Our proposed framework is different from the traditional Bayesian GNN network (Zhang et al., 2018) in that the output of the former are the parameters of node-level Dirichlet distributions ($\boldsymbol{\alpha}$), but the output of the latter are directly node-level class probabilities ($\mathbf{p}$). The conditional probability of $\mathbf{p}$, $\text{Prob}(\mathbf{p}|\mathbf{r}; \boldsymbol{\theta})$, can be obtained by:

$$\text{Prob}(\mathbf{p}|\mathbf{r}; \boldsymbol{\theta}) = \prod_{i=1}^{N} \text{Dir}(\mathbf{p}_i|\boldsymbol{\alpha}_i), \ \boldsymbol{\alpha}_i = f_i(\mathbf{r}; \boldsymbol{\theta}) \tag{3}$$

where the Dirichlet probability function $\text{Dir}(\mathbf{p}_i|\boldsymbol{\alpha}_i)$ is defined by:

$$\text{Dir}(\mathbf{p}_i|\boldsymbol{\alpha}_i) = \frac{\Gamma(S_i)}{\prod_{k=1}^{K} \Gamma(\alpha_{ik})} \prod_{k=1}^{K} p_{ik}^{\alpha_{ik}-1}, \ S_i = \sum_{k=1}^{K} \alpha_{ik} \tag{4}$$

Based on the proposed Bayesian GNN framework, the joint probability of $\mathbf{y}$ conditioned on the input graph $\mathcal{G}$ and the node-level feature matrix $\mathbf{r}$ can be estimated by:

$$\text{Prob}(\mathbf{y}|\mathbf{r}; \mathcal{G}) = \int \int \text{Prob}(\mathbf{y}|\mathbf{p})\text{Prob}(\mathbf{p}|\mathbf{r}; \boldsymbol{\theta})\text{Prob}(\boldsymbol{\theta}|\mathcal{G})d\mathbf{p}d\boldsymbol{\theta}, \tag{5}$$

where $\text{Prob}(\boldsymbol{\theta}|\mathcal{G})$ is the posterior probability of the parameters $\boldsymbol{\theta}$ conditioned on the input graph $\mathcal{G}$, which are estimated in Sections 3.4 and 3.6.

The *aleatoric uncertainty* and the *epistemic uncertainty* can be estimated using the equations described in Section 3.2. The *vacuity* associated with the class probabilities ($\mathbf{p}_i$) of node $i$ can be estimated by: $\text{Vacuity}(\mathbf{p}_i) = \mathbb{E}_{\text{Prob}(\boldsymbol{\theta}|\mathcal{G})}[v_i] = \mathbb{E}_{\text{Prob}(\boldsymbol{\theta}|\mathcal{G})}\left[K/\sum_{k=1}^{K}\alpha_{ik}\right]$. The *dissonance* of node $i$ is estimated as: $\text{Disso.}[\mathbf{p}_i] = \mathbb{E}_{\text{Prob}(\boldsymbol{\theta}|\mathcal{G})}\left[\omega(b_i)\right]$, where $\omega(b_i)$ is defined in Eq. (2).

### 3.4 BAYESIAN INFERENCE WITH DROPOUT

The marginalization in Eq. (5) is generally intractable. A dropout technique is used to obtain an approximate solution and use samples from the posterior distribution of models (Gal & Ghahramani, 2016). Due to this reason, we adopt a dropout technique in (Gal & Ghahramani, 2015) for variational inference in Bayesian CNNs where Bernoulli distributions are assumed over the network's weights. This dropout technique allows us to perform probabilistic inference over our Bayesian DL framework using GNNs. For Bayesian inference, we identify a posterior distribution over the network's weights, given the input graph $\mathcal{G}$ and observed labels $\mathbf{y}_{\mathbb{L}}$ by $\text{Prob}(\boldsymbol{\theta} \mid \mathcal{G})$, where $\boldsymbol{\theta} = \{\mathbf{W}_1, \dots, \mathbf{W}_L, b_1, ..., b_L\}$, $L$ is the total number of layers and $W_i$ refers to the GNN's weight matrices of dimensions $P_i \times P_{i-1}$, and $b_i$ is a bias vector of dimensions $P_i$ for layer $i = 1, \cdots, L$.

Since the posterior distribution is intractable, we use a **variational inference** to learn $q(\boldsymbol{\theta}, \boldsymbol{\gamma})$, a distribution over matrices whose columns are randomly set to zero, approximating the intractable posterior by minimizing the Kullback-Leibler (KL)-divergence between this approximated distribution and the full posterior, which is given by:

$$\min_{\boldsymbol{\gamma}} \text{KL}(q(\boldsymbol{\theta}, \boldsymbol{\gamma})\|\text{Prob}(\boldsymbol{\theta}|\mathcal{G})) \tag{6}$$

where $\boldsymbol{\gamma} = \{\mathbf{M}_1, \dots, \mathbf{M}_L, \mathbf{m}_1, \dots, \mathbf{m}_L\}$ are the variational parameters, where $\mathbf{M}_i \in \mathbb{R}^{P_i \times P_{i-1}}$ and $\mathbf{m}_i \in \mathbb{R}^{P_i}$. We define $\mathbf{W}_i$ in $q(\boldsymbol{\theta}, \boldsymbol{\gamma})$ by:

$$\mathbf{W}_i = \mathbf{M}_i\text{diag}([z_{ij}]_{j=1}^{P_i}), \quad z_{ij} \sim \text{Bernoulli}(d_i) \text{ for } i = 1, \dots, L, j = 1, \dots, P_{i-1} \tag{7}$$

where $\mathbf{d} = \{d_1, \dots, d_L\}$ is the dropout probabilities with $z_{ij}$ of Bernoulli distributed random variables. The binary variable $z_{ij} = 0$ corresponds to unit $j$ in layer $i - 1$ being dropped out as an input to layer $i$. We can obtain the approximate model of the Gaussian process from (Gal & Ghahramani, 2015). The dropout probabilities, $d_i$'s, can be optimized or fixed (Kendall et al., 2015). For simplicity, we fix $d_i$'s in our experiments, as it is beyond the scope of our study. In (Gal & Ghahramani, 2015), the minimization of the cross entropy (or square error) loss function is proven to minimize the KL-divergence (see Eq. (6)). Therefore, training the GNN model with stochastic gradient descent enables learning of an approximated distribution of weights, which provides good explainability of data and prevents overfitting.

For the dropout inference, we performed training a GNN model with dropout before every weight layer and dropout at test time to sample from the approximate posterior (i.e., stochastic forward passes, a.k.a. Monte Carlo dropout; see Eq. (8)). At the test stage, we infer the joint probability Eq. (5) by:

$$\text{Prob}(\mathbf{y}|\mathbf{r}; \mathcal{G}) \approx \frac{1}{M}\sum_{m=1}^{M}\int \text{Prob}(\mathbf{y}|\mathbf{p})\text{Prob}(\mathbf{p}|\mathbf{r}; \boldsymbol{\theta}^{(m)})d\mathbf{p}, \quad \boldsymbol{\theta}^{(m)} \sim q(\boldsymbol{\theta}), \tag{8}$$

which can infer the Dirichlet parameters $\boldsymbol{\alpha}$ as: $\boldsymbol{\alpha} \approx \frac{1}{M}\sum_{m=1}^{M}f(\mathbf{r}, \boldsymbol{\theta}^{(m)}), \quad \boldsymbol{\theta}^{(m)} \sim q(\boldsymbol{\theta})$.

As our model is a generative model to predict Dirichlet distribution parameters, we use a *loss function* to compute its Bayes risk with respect to the sum of squares loss $\|\mathbf{y} - \mathbf{p}\|_2^2$ by:

$$\mathcal{L}(\boldsymbol{\gamma}) = \sum_{i\in\mathbb{L}}\int \|\mathbf{y}_i - \mathbf{p}_i\|_2^2 \cdot \text{Prob}(\mathbf{p}_i|\mathbf{r}; \boldsymbol{\gamma})d\mathbf{p}_i = \sum_{i\in\mathbb{L}}\sum_{j=1}^{K}\left(y_{ij} - \mathbb{E}[p_{ij}]\right)^2 + \text{Var}(p_{ij}) \tag{9}$$

Eq. (9) aims to minimize the prediction error and variance, leading to maximizing classification accuracy of each training node by removing excessive misleading evidence (Sensoy et al., 2018).

### 3.5 Graph-Based Kernel Dirichlet Distribution Estimation

To better learn the Dirichlet distribution from our Bayesian GNN framework, we proposed a Graph-Based Kernel Dirichlet Distribution Estimation (GKDE). The key idea of GKDE is estimating prior Dirichlet distribution parameters for each node based on training nodes (see Figure 1 (b)). And then, we leave prior Dirichlet distribution in the training process to learn two trends: (i) nodes with high vacuity (due to lack of evidence) will be shown far from training nodes; and (ii) nodes with high dissonance (due to conflicting evidence) will be shown in the class boundary.

Based on SL, let each training node represent one evidence for its class label. Denote the contribution of evidence estimation for target node $j$ from node $i$ by $\mathbf{h}(y_i, dis(i,j)) = [h_1, \ldots, h_k, \ldots, h_K] \in [0,1]^K$ and $h_k(y_i, dis(i,j))$ is obtained by:

$$h_k(y_i, dis(i,j)) = \begin{cases} 0 & y_i \neq k \\ \sigma\sqrt{2\pi} \cdot g(dis(i,j)) & y_i = k \end{cases} \qquad (10)$$

where $g(dis(i,j)) = \frac{1}{\sigma\sqrt{2\pi}} e^{-\frac{dis(i,j)^2}{2\sigma^2}}$ is the Gaussian kernel function to estimate the distribution effect between nodes $i$ and $j$, and $dis(i,j)$ means the **node distance** (**shortest path between nodes $i$ and $j$**), and $\sigma$ is the bandwidth parameter. The prior evidence estimation based GKDE: $\hat{e}_j = \sum_{i \in \mathbb{L}} \mathbf{h}(y_i, dis(i,j))$, and the prior Dirichlet distribution $\hat{\boldsymbol{\alpha}}_j = \hat{e}_j + \mathbf{1}$. During training process, we minimize the KL-divergence between model predictions of Dirichlet distribution and prior distribution: $\min \text{KL}[\text{Dir}(\boldsymbol{\alpha}) \| \text{Dir}(\hat{\boldsymbol{\alpha}})]$.

### 3.6 A Teacher Network for Refined Inference

Our key contribution is that the proposed Bayesian GNN model is capable of estimating various uncertainty types to predict existing GNNs. As one of the preferred features, the expected class probabilities generated by our Bayesian GNNs model should be consistent with the predicted class probabilities of the GNN model. In addition, our Bayesian GNN model is *a generative model* and may not necessarily always outperform GNN models (i.e., *discriminative models*) for the task of node classification prediction when uncertainty-based prediction is not fully benefited.

To refine the inference of our proposed model, we leverage the principles of Knowledge Distillation in DL (Hinton et al., 2015). In particular, we consider our proposed model as a distilled model and a deterministic GNN model as a teacher model, as shown in Figure 1 (c). The key idea is to train our proposed model to imitate the outputs of the teacher network on the class probabilities while minimizing the loss function of our proposed model. We observed that the modeling of data uncertainty in our proposed model provides useful information to further improve the accuracy of the deterministic GNN model. Therefore, we consider propagating the useful information back to the teacher model to help train itself.

Let us denote $\text{Prob}(\mathbf{y} \mid \mathbf{r}; \boldsymbol{\beta})$ as the joint probability of class labels via a deterministic GNN model, where $\boldsymbol{\beta}$ refers to model parameters. The probability function $\text{Prob}(\mathbf{y} \mid \mathbf{r}; \boldsymbol{\gamma}, \mathcal{G})$ is estimated based on Eq. (8) using the variational parameters $\boldsymbol{\gamma}$. We measure the closeness between $\text{Prob}(\mathbf{y} \mid \mathbf{r}; \boldsymbol{\beta})$ and $\text{Prob}(\mathbf{y} \mid \mathbf{r}; \boldsymbol{\gamma}, \mathcal{G})$ with KL-divergence to be minimized while minimizing their own loss functions based on the labeled nodes. This leads to solving the following optimization problem:

$$\min_{\boldsymbol{\gamma}, \boldsymbol{\beta}} \mathcal{L}(\boldsymbol{\gamma}) + \mathcal{L}(\boldsymbol{\beta}) + \lambda \cdot \Big( \text{KL}[\text{Prob}(\mathbf{y} \mid \mathbf{r}; \boldsymbol{\gamma}, \mathcal{G}) \parallel \text{Prob}(\mathbf{y} \mid \mathbf{r}; \boldsymbol{\beta})] + \text{KL}[\text{Dir}(\alpha) \| \text{Dir}(\hat{\alpha})] \Big) \qquad (11)$$

where $\mathcal{L}(\beta)$ is the loss function (i.e., cross entropy) of the deterministic GNN model and $\lambda$ is a trade-off parameter. Our inference algorithm using backpropagation is detailed in the Appendix.

## 4 Experiments

In this section, we describe our experimental settings and demonstrate the performance of our proposed model based on semi-supervised node classification. For the performance comparison and analysis of our model and other existing counterparts, we demonstrate and analyze the obtained results in terms of the overall classification accuracy.

### 4.1 Datasets

We use six datasets, including three citation network datasets (Sen et al., 2008) (i.e., Cora, Citeseer, Pubmed) and three new datasets (Shchur et al., 2018) (i.e., Coauthor Physics, Amazon Computer, and Amazon Photo). We summarize the description and experimental setup of the used datasets in Table 1. For all the used datasets, we deal with undirected graphs with 20 training nodes for each

Table 1: Description of datasets and their experimental setup for the node classification prediction.

|  | Cora | Citeseer | Pubmed | Co. Physics | Ama.Computer | Ama.Photo |
|---|---|---|---|---|---|---|
| #Nodes | 2,708 | 3,327 | 19,717 | 34, 493 | 13, 381 | 7, 487 |
| #Edges | 5,429 | 4,732 | 44,338 | 282, 455 | 259, 159 | 126, 530 |
| #Classes | 7 | 6 | 3 | 5 | 10 | 8 |
| #Features | 1,433 | 3,703 | 500 | 8,415 | 767 | 745 |
| #Training nodes | 140 | 120 | 60 | 100 | 200 | 160 |
| #Validation nodes | 500 | 500 | 500 | 500 | 500 | 500 |
| #Test nodes | 1,000 | 1,000 | 1,000 | 1000 | 1,000 | 1000 |

category. We chose the same dataset splits as in (Yang et al., 2016) with an additional validation node set of 500 labeled examples for the hyperparameter obtained from the citation datasets, and followed the same dataset splits in (Shchur et al., 2018) for Coauthor Physics, Amazon Computer, and Amazon Photo datasets, for fair comparison.

## 4.2 COMPARING SCHEMES

We conduct the extensive comparative performance analysis based on our proposed models and a number of other state-of-the-art counterparts. Our proposed Subjective GNN models are: (1) S-GNN (Subjective GNN) with vacuity and dissonance estimation, which outputs subjective opinion (Dirichlet distribution) instead of softmax probability; (2) S-BGNN (Subjective Bayesian GNN), Subjective Graph Neural Network with Bayesian framework with multiple type uncertainty estimation; (3) S-BGNN-T (S-BGNN with Teacher network), help improve the expected class probability estimation; (4) S-BGNN-T-K (S-BGNN-T with GKDE), help improve the Dirichlet distribution estimation. Here we use two popular GNN models: GCN (Kipf & Welling, 2016) and GAT (Veličković et al., 2018).

Our proposed models are compared against a number of the state-of-the-art counterparts. For evaluating three citation datasets (i.e., Cora, Citeseer, Pubmed), we compared our models with: (1) GCN (Kipf & Welling, 2016); (2) GAT (Veličković et al., 2018); (3) non-parametric Bayesian GCNN (BGCNN) (Pal et al., 2019); (4) Bayesian GCN (Zhang et al., 2019); (5) MC-dropout for Bayesian GNN (GCN-Drop, GAT-Drop) (Ryu et al., 2019); (6) skip-gram based graph embeddings (Deep-Walk) (Perozzi et al., 2014); (7) iterative classification algorithm (ICA) (Lu & Getoor, 2003); and (8) Planetoid (Yang et al., 2016). We selected these for the comparison with our models based on (Veličković et al., 2018) for fair comparison with the latest comparable models. Using Coauthor Physics, Amazon Computer, and Amazon Photo, we compared the performance of our models with that of GCN and GAT(Shchur et al., 2018), we can not show S-GAT (S-BGAT) due to memory limited, More details of model setup refer to the Appendix A.

Table 2: Semi-supervised node classification accuracy.

|  | Cora | Citeseer | Pubmed |
|---|---|---|---|
| DeepWalk | 67.2 | 43.2 | 65.3 |
| ICA | 75.1 | 69.1 | 73.9 |
| Planetoid | 75.7 | 64.7 | 77.2 |
| GCN | 81.5 | 70.3 | 79.0 |
| Bayesian GCN | $81.2 \pm 0.8$ | $72.2 \pm 0.6$ | $76.6 \pm 0.7$ |
| BGCNN | $80.3 \pm 0.6$ | $\mathbf{72.6 \pm 0.6}$ | $79.2 \pm 0.5$ |
| GCN-Drop | $81.3 \pm 0.7$ | $70.9 \pm 0.5$ | $79.0 \pm 0.3$ |
| S-GCN | $81.5 \pm 1.0$ | $71.2 \pm 0.6$ | $79.0 \pm 0.2$ |
| S-BGCN | $81.2 \pm 1.0$ | $71.0 \pm 0.6$ | $79.0 \pm 0.2$ |
| S-BGCN-T | $\mathbf{82.2 \pm 0.6}$ | $71.3 \pm 0.6$ | $79.2 \pm 0.4$ |
| S-BGCN-T-K | $82.0 \pm 0.6$ | $71.0 \pm 0.6$ | $\mathbf{79.3 \pm 0.3}$ |
| GAT | $83.0 \pm 0.7$ | $72.5 \pm 0.7$ | $79.0 \pm 0.3$ |
| GAT-Drop | $82.8 \pm 0.8$ | $72.6 \pm 0.7$ | $79.0 \pm 0.3$ |
| S-GAT | $83.0 \pm 0.7$ | $72.6 \pm 0.6$ | $79.0 \pm 0.3$ |
| S-BGAT | $82.9 \pm 0.7$ | $72.4 \pm 0.7$ | $78.9 \pm 0.3$ |
| S-BGAT-T | $83.7 \pm 0.6$ | $\mathbf{73.2 \pm 0.5}$ | $79.1 \pm 0.2$ |
| S-BGAT-T-K | $\mathbf{83.8 \pm 0.7}$ | $73.0 \pm 0.7$ | $\mathbf{79.1 \pm 0.2}$ |
|  | Co.Physics | Ama.Computer | Ama.Photo |
| GAT* | $92.5 \pm 0.9$ | $78.0 \pm 19.0$ | $85.7 \pm 20.3$ |
| GCN* | $92.8 \pm 1.0$ | $82.6 \pm 2.4$ | $91.2 \pm 1.2$ |
| GCN | $93.0 \pm 0.8$ | $79.7 \pm 1.3$ | $91.6 \pm 1.2$ |
| GCN-Drop | $93.0 \pm 0.7$ | $79.6 \pm 1.2$ | $91.3 \pm 1.0$ |
| S-GCN | $93.1 \pm 0.8$ | $78.9 \pm 1.6$ | $90.4 \pm 1.4$ |
| S-BGCN | $93.1 \pm 0.7$ | $78.3 \pm 1.6$ | $90.2 \pm 1.6$ |
| S-BGCN-T | $\mathbf{93.2 \pm 0.8}$ | $\mathbf{84.1 \pm 1.3}$ | $\mathbf{92.3 \pm 1.2}$ |
| S-BGCN-T-K | $93.0 \pm 0.8$ | $84.0 \pm 1.2$ | $92.0 \pm 1.3$ |

GCN* and GAT* are implemented from (Shchur et al., 2018)

## 4.3 EXPERIMENTAL RESULTS & ANALYSIS

In Table 2, we summarized the mean percentage of classification accuracy with a standard deviation of each model compared in this experiment. The results prove that our model achieves the best accuracy result across all datasets except Citeseer. To be specific, our proposed S-BGCN-T is able to improve over GCN by a margin of 0.7%, 1.2%, 0.2%, 0.2%, 4.5% and 0.7% on Cora, Citeseer, Pumbed, Coauthor Physics, Amazon Computer, and Amazon Photo, respectively. In addition, our proposed S-BGAT-T model improves 0.8% for both Cora and Citeseer datasets over GAT. Notice that S-BGNN-T even outperforms S-BGNN particularly on the Cora and Citeseer dataset (i.e., 1% - 1.3% increase). These results prove that the teacher network can prevent overfitting, leading to a further improvement in classification prediction.

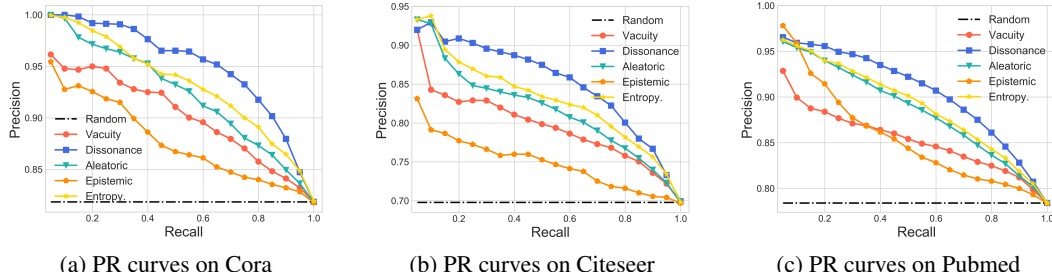

(a) PR curves on Cora      (b) PR curves on Citeseer      (c) PR curves on Pubmed

Figure 2: **PR curves for Node Classification Prediction**: S-BGCN-T-K is used to predict node classification where test nodes are selected based on the low extent of uncertainty. The PR curves of all compared models can be found in the Appendix.

# 5 UNCERTAINTY EXPERIMENT AND ANALYSIS

In Section 4, we showed that our S-BGNN-T improves prediction performance. In this section, we study the effectiveness of prediction based on different types of uncertainty. We studied the different types of uncertainty-aware node classification and out-of-distribution in terms of the area under the ROC (AUROC) and Precision-Recall (AUPR) curves in both experiments as in (Hendrycks & Gimpel, 2016) for three citation network datasets. For the OOD detection, we randomly selected 1-3 categories as OOD categories and trained the models only based on training nodes of the other categories. Due to the space constraint, we summarize the description of datasets and experimental setup for the OOD detection in the Appendix.

To better evaluate our multiple uncertainties, we compare our model with two baseline models: (1) GCN which uses GCN (Kipf & Welling, 2016) with the softmax probability entropy measuring uncertainty; and (2) GCN-Drop, where one of the two uncertainty types (i.e., aleatoric, or epistemic uncertainty) adapts Monte-Carlo Dropout (Gal & Ghahramani, 2016) into the GCN model (Ryu et al., 2019). In the OOD, we also consider Distributional uncertainty (Malinin & Gales, 2018).

## 5.1 QUALITY OF UNCERTAINTY METRICS

In Figure 2, we used S-BGCN-T-K to predict node classification when test nodes are selected based on the lowest uncertainty for a given type. First of all, all uncertainty types show decreasing precision as recall increases. This implies that all uncertainty types are to some extent the indicators of prediction accuracy because low uncertainty increases prediction accuracy. In Figure 2, we can observe almost 100% performance of precision when recall is close to zero on Cora and over 95% on Pubmed. Further, the outperformance of Dissonance uncertainty is obvious among all. This indicates that low uncertainty with few conflicting evidence is the most critical factor to enhance classification prediction accuracy, compared to low extent of other uncertainty types. In addition, although epistemic uncertainty was very low, epistemic uncertainty performs the worst among all. This also indicates that epistemic uncertainty is not necessarily helpful to enhance prediction accuracy in semi-supervised node classification. Lastly, we found that vacuity is not as important as dissonance because accurate prediction is not necessarily dependent upon a large amount of information, but is more affected by less conflicting (or more agreeing) evidence to support a single class.

Table 3: Node classification prediction in AUPR.

| Data | Model | AUPR | | | | |
|---|---|---|---|---|---|---|
| | | Va. | Dis. | Al. | Ep. | En. |
| Cora | S-BGCN-T-K | 90.4 | **95.4** | 92.6 | 88.0 | 93.4 |
| | S-BGCN-T | 89.5 | 94.4 | 91.9 | 85.3 | 92.2 |
| | GCN-Drop. | - | - | 92.7 | 90.0 | 93.6 |
| | GCN | - | - | - | - | 94.1 |
| Citeseer | S-BGCN-T-K | 79.8 | **85.6** | 82.2 | 75.2 | 83.5 |
| | S-BGCN-T | 80.0 | **85.6** | 82.4 | 76.6 | 83.6 |
| | GCN-Drop. | - | - | 82.3 | 77.8 | 83.7 |
| | GCN | - | - | - | - | 83.2 |
| Pubmed | S-BGCN-T-K | 85.6 | **90.9** | 88.9 | 86.0 | 89.3 |
| | S-BGCN-T | 84.8 | 90.2 | 88.8 | 85.1 | 89.3 |
| | GCN-Drop. | - | - | 88.6 | 85.6 | 89.0 |
| | GCN | - | - | - | - | 89.2 |

Va.: Vacuity, Dis.: Dissonance, Al.: Aleatoric, Ep.: Epistemic, En.: Entropy

In Table 3, although all BGCN-T models with the five different uncertainty types do not necessarily outperform all the existing models (i.e., GCN Entropy and variants of GCN-Drop.), the outperformance of Dissonance is fairly impressive. This result confirmed that low uncertainty caused by dissonance is the key to maximize node classification prediction accuracy. When compare S-BGCN-T and S-BGCN-T-K, we found GKDE can only help improve performance a little. To better understand different uncertainty types, we used $t$-SNE (Maaten & Hinton, 2008) to represent the computed

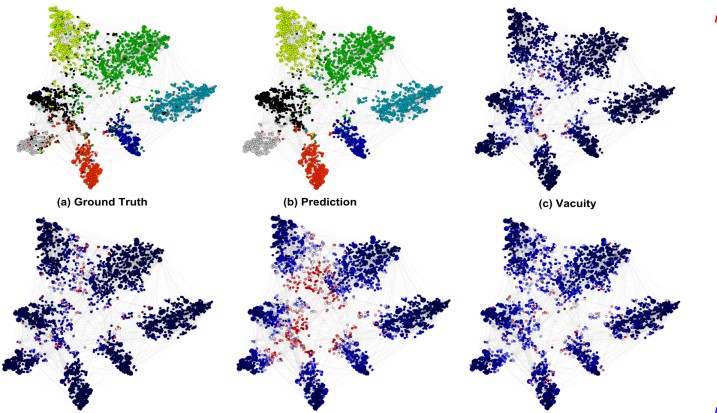

Figure 3: Graph embedding representations of the Cora dataset for classes and the extent of uncertainty: (a) shows the representation of seven different classes; (b) shows our model prediction; and (c)-(f) present the extent of uncertainty for respective uncertainty types, including vacuity, dissonance, aleatoric, epistemic.

feature representations of a pre-trained S-BGCN-T-K model's first hidden layer on the Cora dataset in Figure 3.

## 5.2 OUT-OF-DISTRIBUTION DETECTION

In this section, we discuss how different uncertainty types can prove the performance in the out-of-distribution (ODD) detection. In Table 4, we considered 6 uncertainties with 3 models for our performance comparison. Note that Distributional uncertainty is the the most recent model showing the best performance in the OOD detection. Across the three citation network datasets, particularly S-BGCN-T-K Vacuity showed significantly better performance, strikingly outperforming Distributional uncertainty. Notice that S-BGCN-T-K outperforms S-BGCN-T (i.e., 4% - 7% increase), especially the improvement of vacuity. These result prove that the GKDE can improve the Dirichlet distribution estimation, leading to a better uncertainty estimation.

Table 4: AUROC and AUPR for the OOD detection.

| Data | Model | AUROC | | | | | | AUPR | | | | | |
|---|---|---|---|---|---|---|---|---|---|---|---|---|---|
| | | Va. | Dis. | Al. | Ep. | D.En. | En. | Va. | Dis. | Al. | Ep. | D.En. | En. |
| Cora | S-BGCN-T-K | **87.6** | 75.5 | 85.5 | 70.8 | 85.1 | 84.8 | **78.4** | 49.0 | 75.3 | 44.5 | 73.8 | 73.1 |
| | S-BGCN-T | 84.5 | 81.2 | 83.5 | 71.8 | 84.1 | 83.5 | 74.4 | 53.4 | 75.8 | 46.8 | 70.8 | 71.7 |
| | GCN-Drop. | - | - | 81.9 | 70.5 | - | 80.9 | - | - | 69.7 | 44.2 | - | 67.2 |
| | GCN | - | - | - | - | - | 80.7 | - | - | - | - | - | 66.9 |
| Citeseer | S-BGCN-T-K | **84.8** | 55.2 | 78.4 | 55.1 | 79.1 | 74.0 | **86.8** | 54.1 | 80.8 | 55.8 | 81.3 | 74.0 |
| | S-BGCN-T | 78.6 | 59.6 | 73.9 | 56.1 | 75.1 | 69.3 | 79.8 | 57.4 | 76.4 | 57.8 | 78.3 | 69.3 |
| | GCN-Drop. | - | - | 72.3 | 61.4 | - | 70.6 | - | - | 73.5 | 60.8 | - | 70.0 |
| | GCN | - | - | - | - | - | 70.8 | - | - | - | - | - | 70.2 |
| Pubmed | S-BGCN-T-K | **74.6** | 67.9 | 71.8 | 59.2 | 69.7 | 72.2 | **69.6** | 52.9 | 63.6 | 44.0 | 64.8 | 56.5 |
| | S-BGCN-T | 71.8 | 68.6 | 70.0 | 60.1 | 68.0 | 70.8 | 65.7 | 53.9 | 61.8 | 46.0 | 62.9 | 55.1 |
| | GCN-Drop. | - | - | 68.7 | 60.8 | - | 66.7 | - | - | 59.7 | 46.7 | - | 54.8 |
| | GCN | - | - | - | - | - | 68.3 | - | - | - | - | - | 55.3 |

Va.: Vacuity, Dis.: Dissonance, Al.: Aleatoric, Ep.: Epistemic, D.En.: Differential Entropy, En.: Entropy

## 6 CONCLUSION

In this work, we proposed a Subjective Bayesian GNNs framework for uncertainty-aware semi-supervised node classification and out-of-distribution (OOD) detection for GNNs. Our proposed framework provides an effective, efficient way of predicting node classification and detecting OOD considering multiple uncertainty types. We leveraged the estimation of various types of uncertainty from both DL and evidence/belief theory domains. In addition, We leveraged the Teacher network to help refine the classification probability and GKDE to accurctly estimate Dirichlet distribution.

The **key findings** from this study include:

- For the overall classification prediction, our proposed S-BGNN-T outperformed the competitive baselines on most datasets. The key role to improve the accuracy is Teacher Network.
- For the node classification prediction considering various uncertainty types, we found that dissonance (i.e., uncertainty derived from conflicting evidence) played a significant role to improve classification prediction accuracy.
- For the OOD detection, vacuity uncertainty played a key role when S-BGCN-T-K is used to detect OOD. This means that less information and/or more randomness (or less predictability) enables detecting OOD more effectively. More impressively, GKDE can indeed help to estimate Dirichlet distribution accurately so that enhance the vacuity performance. Also vacuity outperformed the most recent counterpart, Distributional uncertainty.

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

# A APPENDIX

## A.1 SOURCE CODE

For review purpose, the source code and datasets are accessible at `https://www.dropbox.com/sh/cs5gs2i1umdx4b6/AAC-r_EYRw9lryk95giqW8-Fa?dl=0`

### DESCRIPTION OF DATASETS

**Cora, Citeseer, and Pubmed** (Sen et al., 2008): These are citation network datasets, where a network is a directed network where a node represents a document and an edge is a citation link, meaning that there exists an edge when $A$ document cites $B$ document, or vice-versa with a direction. Each node's feature vector contains a bag-of-words representation of a document. For simplicity, we don't discriminate the direction of links and treat citation links as undirected edges and construct a binary, symmetric adjacency matrix $\mathbf{A}$. Each node is labeled with the class to which it belongs.

**Coauthor Physics, Amazon Computers, and Amazon Photo** (Shchur et al., 2018): Coauthor Physics is the dataset for co-authorship graphs based on the Microsoft Academic Graph from the KDD Cup 2016 Challenge[1]. In the graphs, a node is an author and an edge exists when two authors co-author a paper. A node's features represent the keywords of its papers and the node's class label indicates its most active field of study. Amazon Computers and Amazon Photo are the segments of an Amazon co-purchase graph (McAuley et al., 2015), where a node is a good (i.e., product), an edge exists when two goods are frequently bought together. A node's features are bag-of-words representation of product reviews and the node's class label is the product category.

In semi-supervised node classification task, the training and test nodes are selected based on (Sen et al., 2008) for the citation network datasets and are randomly selected for the Coauthor Physics, Amazon Computers, and Amazon Photo.

## A.2 EXPERIMENTAL SETUP FOR OUT-OF-DISTRIBUTION (OOD) DETECTION

For OOD detection, we summarize the experiment setup for the use of the three citation network datasets (i.e., Cora, Citeseer, and Pubmed) in Table 5. In this setting, we still focus on the semi-supervised node classification task, but only part of node categories are not using for training. Hence, we suppose that our model only outputs partial categories (as we don't know the OOD category). For example, Cora dataset, we train the model with 80 nodes (20 nodes for each category) with the predictions of 4 categories. Positive ratio is the ratio of out-of-distribution nodes among on all test nodes.

| Dataset | Cora | Citeseer | Pubmed |
|---|---|---|---|
| **#Number of training categories** | 4 | 3 | 2 |
| **#Training nodes** | 80 | 60 | 40 |
| **#Test nodes** | 1000 | 1000 | 1000 |
| **#Positive ratio** | 38% | 55% | 40.4% |

Table 5: Description of datasets and their experimental setup for the OOD detection.

## A.3 CALCULATION OF AUPR AND AUROC

For the calculation of precision, recall, TPR, and FPR, we select a certain $\phi$ % of nodes out of test nodes to label them as positive (correct) based on the extent of uncertainty, the lowest uncertainty for classification prediction and the highest uncertainty for OOD detection. And the remaining test nodes (i.e., $100 - \phi$ %) are labeled as negative. Each test node's prediction is checked with its ground truth to derive AUPR and AUROC.

---

[1]KDD Cup 2016 Dataset: Online Available at `https://kddcup2016.azurewebsites.net/`

TIME COMPLEXITY ANALYZE

BGCN has a similar time complexity with GCN while BGCN-T has the double complexity of GCN. In the revised paper, we will add a table showing Big-O complexity of all schemes considered. For a given network where $|\mathbb{V}|$ is the number of nodes, $|\mathbb{E}|$ is the number of edges, $C$ is the number of dimensions of the input feature vector for every node, and $F$ is the number of features for the output layer, the complexity of the compared schemes are: $O(|\mathbb{E}|CF)$ for GCN, $O(|\mathbb{E}|CF)$ for S-BCGN, $O(2|\mathbb{E}|CF)$ for S-BCGB-T and S-BCGB-T-K, $O(|\mathbb{V}|CF+|\mathbb{E}|F)$ for GAT, and $O(2|\mathbb{V}|CF+2|\mathbb{E}|F)$ for S-BGAT-T and S-BGAT-T-K.

## A.4 MODEL SETUPS FOR SEMI-SUPERVISED NODE CLASSIFICATION

Our models are initialized using Glorot initialization (Glorot & Bengio, 2010) and trained to minimize loss using the Adam SGD optimizer (Kingma & Ba, 2014). As our proposed models (i.e., S-BGCN-T-K, S-BGAT-T-K) need a discriminative model to refine inference, we use standard GCN and GAT models as teacher networks for S-BGCN-T-K and S-BGAT-T-K, respectively. For the S-BGCN-T-K model, we use the *early stopping strategy* (Shchur et al., 2018) on Coauthor Physics, Amazon Computer and Amazon Photo datasets while *non-early stopping strategy* is used in citation datasets (i.e., Cora, Citeseer and Pubmed). We set bandwidth $\sigma = 1$ for all datasets in GKDE, and set trade off parameters $\lambda = \min(1, t/200)$ (where $t$ is the index of a current training epoch); and other hyperparameter configurations are summarized in Table 6. The S-BGAT-T-K model has two dropout probabilities, which are a dropout on features and a dropout on attention coefficients, as showed in Table 7. We changed the dropout on attention coefficients to 0.4 at the test stage and set trade off parameters $\lambda = \min(1, t/50)$, using the same early stopping strategy (Veličković et al., 2018). **Note that** lack of memory (we used one Titan X GPU, 12 GB memory), we could not obtain the result for GAT (also S-BGAT) on Coauthor Physics, Amazon Computer and Amazon Photo, which are very dense datasets.

For semi-supervised node classification, we use 50 random weight initialization for our models on Citation network datasets. For Coauthor Physics, Amazon Computer and Amazon Photo datasets, we report the result based on 10 random train/validation/test splits. In both effect of uncertainty on classification prediction accuracy and the OOD detection, we report the AUPR and AUROC results in percent averaged over 50 times of randomly chosen 1000 test nodes in all of test sets (except training or validation set) for all models tested on the citation datasets. For S-BGCN-T-K model in these tasks, we use the same hyperparameter configurations as in Table 6, except S-BGCN-T-K Epistemic using 20,000 epochs to obtain the best result. For baseline models, GCN-Drop. models use the same hyperparameters as in Table 6 to achieve the best performance, also using 20,000 training epochs for GCN-Drop. Epistemic. GCN Entropy uses the same hyperparameter configurations in (Kipf & Welling, 2016).

|  | **Cora** | **Citeseer** | **Pubmed** | **Co.Physics** | **Ama.Computer** | **Ama.Photo** |
|---|---|---|---|---|---|---|
| **Hidden units** | 16 | 16 | 16 | 64 | 64 | 64 |
| **Learning rate** | 0.01 | 0.01 | 0.01 | 0.01 | 0.01 | 0.01 |
| **Dropout** | 0.5 | 0.5 | 0.5 | 0.1 | 0.2 | 0.2 |
| $L_2$ **reg.strength** | 0.0005 | 0.0005 | 0.0005 | 0.001 | 0.0001 | 0.0001 |
| **Monte-Carlo samples** | 500 | 500 | 500 | 100 | 100 | 100 |
| **Max epoch** | 200 | 200 | 200 | 100000 | 100000 | 100000 |

Table 6: Hyperparameter configurations of S-BGCN-T-K model.

|                      | Cora      | Citeseer  | Pubmed    |
|----------------------|-----------|-----------|-----------|
| **Hidden units**     | 64        | 64        | 64        |
| **Learning rate**    | 0.01      | 0.01      | 0.01      |
| **Dropout**          | 0.6/0.6   | 0.6/0.6   | 0.6/0.6   |
| $L_2$ **reg.strength** | 0.0005  | 0.0005    | 0.001     |
| **Monte-Carlo samples** | 100    | 100       | 100       |
| **Max epoch**        | 100000    | 100000    | 100000    |

Table 7: Hyper-parameters of S-BGAT-T-K model.

## A.5 ALGORITHM FOR OUR ALGORITHM

---

**Algorithm 1:** S-BGNN-T-K with jointly train for Teacher network

---

**Input:** $\mathbb{G} = (\mathbb{V}, \mathbb{E}, \mathbf{r})$ and $\mathbf{y}_\mathbb{L}$

**Output:** $\mathbf{p}_{\mathbb{V}\backslash\mathbb{L}}, \mathbf{u}_{\mathbb{V}\backslash\mathbb{L}}$

1 $\ell = 0$;
2 Set hyper-parameters;
3 Initialize the parameters $\gamma, \beta$;
4 Calculate the prior Dirichlet distribution $\text{Dir}(\hat{\alpha})$;
5 **repeat**
6     Forward pass to compute $\boldsymbol{\alpha}, \text{Prob}(\mathbf{p}_i|\mathbf{r}; \mathcal{G}), \text{Prob}(\mathbf{y}_i|\mathbf{r}; \beta)$ for $i \in \mathbb{V}$;
7     Compute joint probability $\text{Prob}(\mathbf{y}|\mathbf{r}; \mathcal{G}), \text{Prob}(\mathbf{y}|\mathbf{r}; \beta)$;
8     Backward pass via the chain-rule the calculate the sub-gradient gradient: $g^{(\ell)} = \nabla_\Theta \mathcal{L}(\Theta)$
9     Update parameters using step size $\eta$ via $\Theta^{(\ell+1)} = \Theta^{(\ell)} - \eta \cdot g^{(\ell)}$
10     $\ell = \ell + 1$;
11 **until** *convergence*
12 Calculate $\mathbf{p}_{\mathbb{V}\backslash\mathbb{L}}, \mathbf{u}_{\mathbb{V}\backslash\mathbb{L}}$
13 **return** $p_{\mathbb{V}\backslash\mathbb{L}}, u_{\mathbb{V}\backslash\mathbb{L}}$

---

**Algorithm 2:** S-BGNN-T-K with pre-train for Teacher network

---

**Input:** $\mathbb{G} = (\mathbb{V}, \mathbb{E}, \mathbf{r})$ and $\mathbf{y}_\mathbb{L}$

**Output:** $\mathbf{p}_{\mathbb{V}\backslash\mathbb{L}}, \mathbf{u}_{\mathbb{V}\backslash\mathbb{L}}$

1 $\ell = 0$;
2 Set hyper-parameters;
3 Initialize the parameters $\gamma, \beta$;
4 Calculate the prior Dirichlet distribution $\text{Dir}(\hat{\alpha})$;
5 Pre-train the Teacher Network to get $\text{Prob}(\mathbf{y}|\mathbf{r}; \beta)$
6 **repeat**
7     Forward pass to compute $\boldsymbol{\alpha}, \text{Prob}(\mathbf{p}_i|\mathbf{r}; \mathcal{G})$ for $i \in \mathbb{V}$;
8     Compute joint probability $\text{Prob}(\mathbf{y}|\mathbf{r}; \mathcal{G})$;
9     Backward pass via the chain-rule the calculate the sub-gradient gradient: $g^{(\ell)} = \nabla_\Theta \mathcal{L}(\Theta)$
10     Update parameters using step size $\eta$ via $\Theta^{(\ell+1)} = \Theta^{(\ell)} - \eta \cdot g^{(\ell)}$
11     $\ell = \ell + 1$;
12 **until** *convergence*
13 Calculate $\mathbf{p}_{\mathbb{V}\backslash\mathbb{L}}, \mathbf{u}_{\mathbb{V}\backslash\mathbb{L}}$
14 **return** $p_{\mathbb{V}\backslash\mathbb{L}}, u_{\mathbb{V}\backslash\mathbb{L}}$

---

## B ADDITIONAL EXPERIMENT RESULTS

Further experiment have been run in addition to the uncertainty analysis in section 5. First, we show the ablation experiment for each compents we proposed. Second, we show more uncertainty visualization result in network node classification for Citeseer dataset. To better understand the performance of uncertainty quality clearly for each uncertainty, we show the AUROC and AUPR curves for all models and uncertainties.

## B.1 ABLATION EXPERIMENTS

We conducted an additional experiemnt to ensure the benefit of the teacher network. We anticipate that the graph kernel prior will improve the estimation accuracy of Dirichlet distribution. However, due to the space constraint, we didn't show the classification results without using the graph kernel prior. In the revised version, we added a detailed ablation study in the revised paper in order to clearly demonstrate the contribution of the key technical components, including a teacher Network, Graph kernel Dirichlet Estimation (GKDE) and subjective Bayesian framework. The key findings obtained from this experiment are: (1) The teacher Network can further improve node classification accuracy (i.e., 0.2% - 1.5% increase, as shown in Table 2); and (2) GKDE (graph kernel prior) using the uncertainty estimates can enhance OOD detection (i.e., 4% - 7% increase, as shown in Table 9).

Table 8: Ablation experiment for Node classification prediction in AUPR.

| Data | Model | AUPR | | | | |
|------|-------|------|------|------|------|------|
| | | Va. | Dis. | Al. | Ep. | En. |
| Cora | S-BGCN-T-K | 90.4 | **95.4** | 92.6 | 88.0 | 93.4 |
| | S-BGCN-T | 89.5 | 94.4 | 91.9 | 85.3 | 92.2 |
| | S-BGCN. | 89.4 | 94.3 | 91.6 | 85.3 | 92.0 |
| | S-GCN | 89.4 | 94.4 | - | - | - |
| Citeseer | S-BGCN-T-K | 79.8 | **85.6** | 82.2 | 75.2 | 83.5 |
| | S-BGCN-T | 80.0 | **85.6** | 82.4 | 76.6 | 83.6 |
| | S-BGCN. | 79.7 | 85.2 | 82.2 | 76.2 | 83.2 |
| | S-GCN | 79.8 | 85.3 | - | - | - |
| Pubmed | S-BGCN-T-K | 85.6 | **90.9** | 88.9 | 86.0 | 89.3 |
| | S-BGCN-T | 84.8 | 90.2 | 88.8 | 85.1 | 89.3 |
| | S-BGCN. | 84.4 | 90.1 | 88.5 | 85.1 | 89.0 |
| | S-GCN | 84.6 | 90.2 | - | - | - |

Va.: Vacuity, Dis.: Dissonance, Al.: Aleatoric, Ep.: Epistemic, En.: Entropy

Table 9: Ablation exepriemnt on AUROC and AUPR for the OOD detection.

| Data | Model | AUROC | | | | | | AUPR | | | | | |
|------|-------|------|------|------|------|------|------|------|------|------|------|------|------|
| | | Va. | Dis. | Al. | Ep. | D.En. | En. | Va. | Dis. | Al. | Ep. | D.En. | En. |
| Cora | S-BGCN-T-K | **87.6** | 75.5 | 85.5 | 70.8 | 85.1 | 84.8 | **78.4** | 49.0 | 75.3 | 44.5 | 73.8 | 73.1 |
| | S-BGCN-T | 84.5 | 81.2 | 83.5 | 71.8 | 84.1 | 83.5 | 74.4 | 53.4 | 75.8 | 46.8 | 70.8 | 71.7 |
| | S-BGCN | 84.1 | 81.4 | 83.3 | 71.9 | 84.0 | 83.3 | 74.1 | 53.8 | 75.4 | 47.0 | 70.3 | 71.6 |
| | S-GCN | 84.2 | 81.2 | - | - | 83.9 | - | 74.2 | 53.9 | - | - | 70.6 | - |
| Citeseer | S-BGCN-T-K | **84.8** | 55.2 | 78.4 | 55.1 | 79.1 | 74.0 | **86.8** | 54.1 | 80.8 | 55.8 | 81.3 | 74.0 |
| | S-BGCN-T | 78.6 | 59.6 | 73.9 | 56.1 | 75.1 | 69.3 | 79.8 | 57.4 | 76.4 | 57.8 | 78.3 | 69.3 |
| | S-BGCN | 78.5 | 59.7 | 73.2 | 56.5 | 75.4 | 69.1 | 79.6 | 57.8 | 76.0 | 57.9 | 78.1 | 69.0 |
| | S-GCN | 78.6 | 60.0 | - | - | 75.7 | - | 79.9 | 57.8 | - | - | 78.4 | - |
| Pubmed | S-BGCN-T-K | **74.6** | 67.9 | 71.8 | 59.2 | 69.7 | 72.2 | **69.6** | 52.9 | 63.6 | 44.0 | 64.8 | 56.5 |
| | S-BGCN-T | 71.8 | 68.6 | 70.0 | 60.1 | 68.0 | 70.8 | 65.7 | 53.9 | 61.8 | 46.0 | 62.9 | 55.1 |
| | S-BGCN | 71.9 | 68.3 | 70.2 | 60.01 | 68.2 | 70.5 | 65.9 | 53.6 | 61.9 | 46.4 | 62.6 | 55.0 |
| | S-GCN | 71.7 | 68.8 | - | - | 68.8 | - | 65.8 | 53.9 | - | - | 63.0 | - |

Va.: Vacuity, Dis.: Dissonance, Al.: Aleatoric, Ep.: Epistemic, D.En.: Differential Entropy, En.: Entropy

## B.2 GRAPH EMBEDDING REPRESENTATIONS OF DIFFERENT UNCERTAINTY TYPES

To better understand different uncertainty types, we used $t$-SNE ($t$-Distributed Stochastic Neighbor Embedding (Maaten & Hinton, 2008)) to represent the computed feature representations of a pre-trained BGCN-T model's first hidden layer on the Citeseer dataset.

**Six Classes on Citeseer Dataset**: In Figure 4 (a), a node's color denotes a class on the Citeseer dataset where 6 different classes are shown in different colors. Figure 4 (b) is our prediction result.

For Figures 4 (c)-(f), the extent of uncertainty is presented where a blue color refers to the lowest uncertainty (i.e., minimum uncertainty) while a red color indicates the highest uncertainty (i.e., maximum uncertainty) based on the presented color bar. To examine the trends of the extent of uncertainty depending on either training nodes or test nodes, we draw training nodes as bigger circles than test nodes. Overall we notice that most training nodes (shown as bigger circles) have low uncertainty (i.e., blue), which is reasonable because the training nodes are the ones that are already observed. Now we discuss the extent of uncertainty under each uncertainty type.

**Vacuity**: In Figure 4 (c), although most training nodes show low uncertainty, we observe majority of test nodes in the mid cluster show high uncertainty as appeared in red.

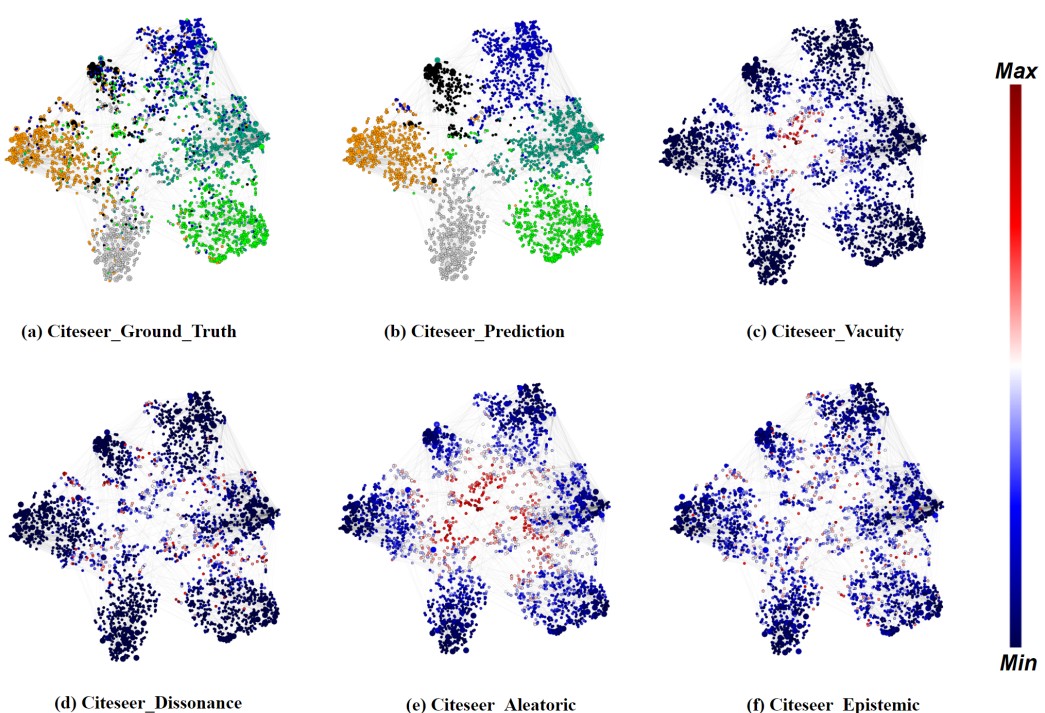

Figure 4: Graph embedding representations of the Citeseer dataset for classes and the extent of uncertainty: (a) shows the representation of seven different classes, (b) shows our model prediction and (c)-(f) present the extent of uncertainty for respective uncertainty types, including vacuity, dissonance, and aleatoric uncertainty, respectively.

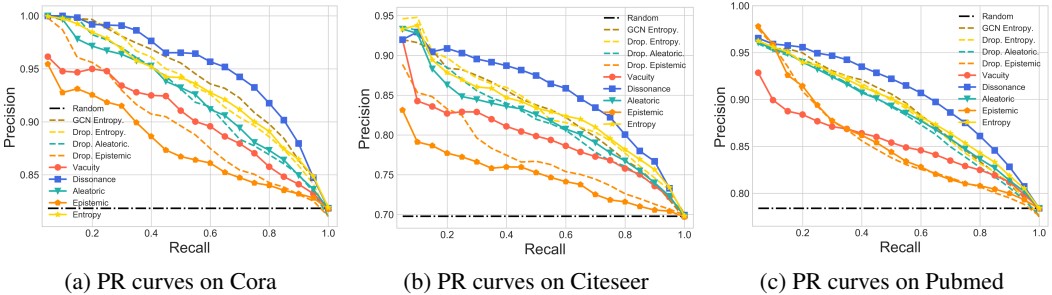

Figure 5: PR curves of node classification prediction for all models and uncertainties.

**Dissonance**: In Figure 4 (d), similar to vacuity, training nodes have low uncertainty. But unlike vacuity, test nodes are much less uncertain. Recall that dissonance represents the degree of conflicting evidence (i.e., discrepancy between each class probability). However, in this dataset, we observe a fairly low level of dissonance and the obvious outperformance of Dissonance in node classification prediction.

**Aleatoric uncertainty**: In Figure 4 (e), a lot of nodes show high uncertainty with larger than 0.5 except a small amount of training nodes with low uncertainty. High aleatoric uncertainty positively affects, showing high performance in OOD detection.

**Epistemic uncertainty**: In Figure 4 (f), most nodes show very low epistemic uncertainty because uncertainty derived from model parameters can disappear as they are trained well. Therefore, non-distinctive low uncertainty for most nodes do not help much to select good test nodes to improve performance in node classification.

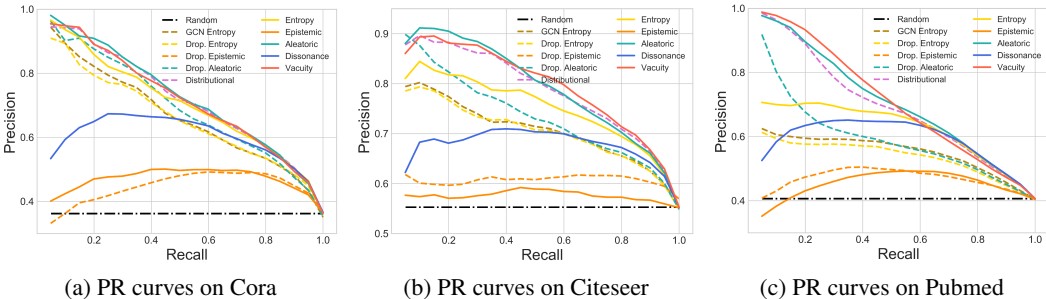

(a) PR curves on Cora      (b) PR curves on Citeseer      (c) PR curves on Pubmed

Figure 6: PR cuves of OOD detection for all models and uncertainties.

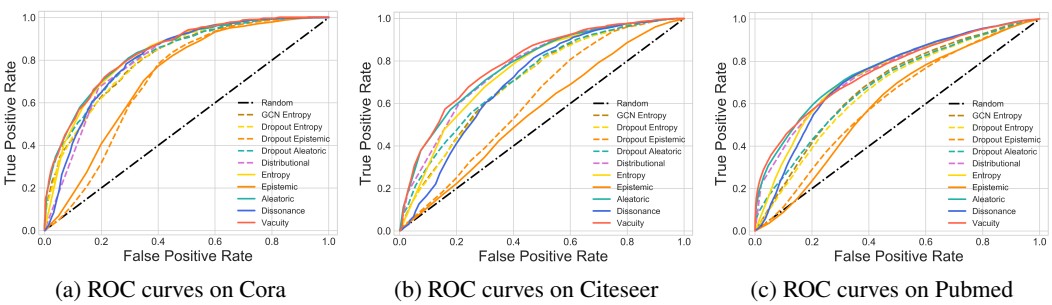

(a) ROC curves on Cora      (b) ROC curves on Citeseer      (c) ROC curves on Pubmed

Figure 7: ROC curves of OOD detection for all models and uncertainties.

### B.3 PR AND ROC CURVES

**AUPRC for the OOD Detection**: Figure 6 shows the AUPRC for the OOD detection when S-BGCN-T-K is used to detect OOD in which test nodes are considered based on their high uncertainty level, given a different uncertainty type, such as vacuity, dissonance, aleatoric, epistemic, or entropy (or total uncertainty). Also to check the performance of the proposed models with a baseline model, we added S-BGCN-T-K with test nodes randomly selected (i.e., Random).

Obviously, in Random baseline, precision was not sensitive to increasing recall while in S-BGCN-T-K (with test nodes being selected based on high uncertainty) precision decreases as recall increases. But although most S-BGCN-T-K models with various uncertainty types used to select test nodes shows sensitive precision to increasing recall (i.e., proving uncertainty being an indicator of improving OOD detection), S-BGCN-T-K Epistemic even performed worse than the baseline (i.e., S-BGCN-T-K Random). This is because epistemic uncertainty cannot distinguish the flat Dirichlet distribution ($\alpha = (1, \ldots, 1)$) from sharp Dirichlet distribution ($\alpha = (10, \ldots, 10)$), which leads to no effective selection of test nodes for improving the performance in OOD detection. In addition, unlike AUPR in node classification prediction, which showed the best performance in S-BGCN-T-K Dissonance (see Figure 5), S-BGCN-T-K Dissonance showed the second worst performance among the proposed S-BGCN-T-K models with other uncertainty types. This means that less conflicting information does not help OOD detection. On the other hand, overall we observe Vacuity performs the best among all while S-BGCN-T-K Entropy also performs fairly well as the third best. From this finding, we can claim that to improve OOD detection, more randomness with high aleatoric uncertainty and less information with high vacuity can help boost the accuracy of the OOD detection. Although the uncertainty level observed from aleatoric uncertainty and entropy was quite similar, the performance in OOD detection is not necessarily similar, as shown in Figures 6 (b) and (c) on Citeseer and Pubmed. The reason is that BCGN-T Aleatoric provides test nodes with more distinctive uncertainty levels while BCGN-T Entropy doesn't. This is because BCGN-T Entropy combines the aleatoric and epistemic uncertainty where epistemic uncertainty is mostly highly low, ultimately leading to poor distinctions of nodes based on different uncertainty levels.

**AUROC for the OOD Detection**: First, we investigated the performance of our proposed S-BGCN-T-K models when test nodes are selected based on seven different criteria (i.e., 6 uncertainties and random). Like AUPR in Figure 5, based on S-BGCN-T-K, we considered a baseline by selecting

test nodes randomly while five different uncertainty types are used to select test nodes based on the order of high uncertainty. For AUROC in Figure 7, we observed much better performance in most S-BGCN-T-K models with all uncertainty types except epistemic uncertainty. Although epistemic uncertainty is known to be effective to improve OOD detection (Kendall & Gal, 2017) in computer vision applications, our result showed fairly poor performance compared to the case other uncertainty types are used. This is because our experiment is conducted with a very small of training nodes (i.e., 3% on Cora, 2% on Citeseer, 0.2% on Pubmed) which is highly challenging to observe high performance particularly with epistemic uncertainty. Recall that we used 200 epochs to train nodes for all models except BCGN-T Epistemic which was trained with 20,000 epochs. In this experiment, even S-BGCN-T-K Vacuity performed the best although S-BGCN-T-K Dissonance, S-BGCN-T-K Aleatoric, or S-BGCN-T-K Entropy performs comparably. But on Citeseer and Pubmed datasets, we also observed relatively low performance with S-BGCN-T-K Dissonance. This finding is also well aligned with what we observed in Table 4 (in paper). S-BGCN-T-K Vacuity performs the best on all three datesets. Obviously S-BGCN-T-K Vacuity outperform S-BGCN-T-K Distributional in OOD detection.

### B.4 ANALYZE FOR EPISTEMIC IN OOD DETECTION

In OOD detection, epistemic uncertainty performed the worst because it cannot distinguish the flat Dirichlet distribution ($\alpha = (1, \ldots, 1)$) from sharp Dirichlet distribution ($\alpha = (10, \ldots, 10)$), resulting poor performance in OOD detection. Unlike AUPR in node classification prediction with outperformance in S-BGCN-T-K Dissonance (see Figure 2), S-BGCN-T-K Dissonance showed the second worst performance among the proposed S-BGCN-T-K models with other uncertainty types. This implies that less conclusive belief mass does not help OOD detection.

Although epistemic uncertainty is known to be effective to improve OOD detection (Kendall & Gal, 2017) in computer vision applications, our result showed fairly poor performance compared to the case other uncertainty types are used. This is because our experiment is conducted with a very small of training nodes (i.e., 3% on Cora, 2% on Citeseer, 0.2% on Pubmed) which is highly challenging to observe high performance particularly with epistemic uncertainty.

## C DERIVATIONS FOR UNCERTAINTY MEASURES AND KL DIVERGENCE

This appendix provides the derivations and shows how calculate the uncertainty measures discussed in section 3 for BGCN. Additionally, it describes how to calculate the joint probability, Dirichlet parameters and KL-divergence between $\text{Prob}(\mathbf{y}|\mathbf{r}; \boldsymbol{\beta})$ and $\text{Prob}(\mathbf{y}|\mathbf{r}; \boldsymbol{\gamma}, \mathcal{G})$.

### C.1 UNCERTAINTY MEASURES

**Vacuity** uncertainty of Bayesian Graph neural networks for node $i$:

$$
\begin{aligned}
\text{Vacuity}[\mathbf{p}_i] &= \mathbb{E}_{\text{Prob}(\boldsymbol{\theta}|\mathcal{G})}[v_i] \\
&= \mathbb{E}_{\text{Prob}(\boldsymbol{\theta}|\mathcal{G})}\Big[K / \sum_{k=1}^{K} \alpha_{ik}\Big] \\
&\approx \mathbb{E}_{q(\boldsymbol{\theta})}\Big[K / \sum_{k=1}^{K} \alpha_{ik}\Big] \\
&\approx \frac{1}{M} \sum_{m=1}^{M} \Big[K / \sum_{k=1}^{K} \alpha_{ik}^{(m)}\Big], \quad \boldsymbol{\alpha}^{(m)} = f(\mathbf{r}, \boldsymbol{\theta}^{(m)}), \quad \boldsymbol{\theta}^{(m)} \sim q(\boldsymbol{\theta})
\end{aligned}
$$

**Dissonance** uncertainty of Bayesian Graph neural networks for node $i$:

$$
\begin{aligned}
\text{Disso.}[\mathbf{p}_i] &= \mathbb{E}_{\text{Prob}(\boldsymbol{\theta}|\mathcal{G})}\Big[\omega(b_i)\Big] \\
&\approx \mathbb{E}_{q(\boldsymbol{\theta})}\Big[\omega(b_i)\Big] \\
&\approx \frac{1}{M}\sum_{m=1}^{M}\Big[\omega(b_i)\Big], \quad \boldsymbol{\theta}^{(m)} \sim q(\boldsymbol{\theta})
\end{aligned}
$$

and

$$
\omega(b_i) = \sum_{k=1}^{K}\Big(\frac{b_{ik}\sum_{j=1,j\neq k}^{K}b_{ij}\text{Bal}(b_{ij},b_{ik})}{\sum_{j=1,j\neq k}^{K}b_{ij}}\Big),
$$

where the relative mass balance between a pair of belief masses $b_{ij}$ and $b_{ik}$ is expressed by $\text{Bal}(b_{ij},b_{ik}) = 1 - |b_{ij}-b_{ik}|/(b_{ij}+b_{ik})$.

**Aleatoric** uncertainty of Bayesian Graph neural networks for node $i$, followed (Malinin & Gales, 2018):

$$
\begin{aligned}
\text{Aleatoric}[\mathbf{p}_i] &= \mathbb{E}_{\text{Prob}(\boldsymbol{\theta}|\mathcal{G})}\big[\mathcal{H}(\mathbf{y}_i|\mathbf{r};\boldsymbol{\theta})\big] \\
&\approx \mathbb{E}_{q(\boldsymbol{\theta})}\big[\mathcal{H}(\mathbf{y}_i|\mathbf{r};\boldsymbol{\theta})\big] \\
&\approx \frac{1}{M}\sum_{m=1}^{M}\mathcal{H}\big[(\mathbf{y}_i|\mathbf{r};\boldsymbol{\theta}^{(m)})\big], \quad \boldsymbol{\theta}^{(m)} \sim q(\boldsymbol{\theta}) \\
&\approx \frac{1}{M}\sum_{m=1}^{M}\sum_{j=1}^{K}\text{Prob}(y_i=j|\mathbf{r};\boldsymbol{\theta}^{(m)})\log\Big(\text{Prob}(y_i=j|\mathbf{r};\boldsymbol{\theta}^{(m)})\Big), \quad \boldsymbol{\theta}^{(m)} \sim q(\boldsymbol{\theta})
\end{aligned}
$$

**Epistemic** uncertainty of Bayesian Graph neural networks for node $i$, followed (Gal, 2016):

$$
\begin{aligned}
\text{Epistemic}[\mathbf{p}_i] &= \mathcal{H}\big[\mathbb{E}_{\text{Prob}(\boldsymbol{\theta}|\mathcal{G})}[(\mathbf{y}_i|\mathbf{r};\boldsymbol{\theta})]\big] - \mathbb{E}_{\text{Prob}(\boldsymbol{\theta}|\mathcal{G})}\big[\mathcal{H}(\mathbf{y}_i|\mathbf{r};\boldsymbol{\theta})\big] \\
&\approx \mathcal{H}\big[\mathbb{E}_{q(\boldsymbol{\theta})}[(\mathbf{y}_i|\mathbf{r};\boldsymbol{\theta})]\big] - \mathbb{E}_{q(\boldsymbol{\theta})}\big[\mathcal{H}(\mathbf{y}_i|\mathbf{r};\boldsymbol{\theta})\big] \\
&\approx \mathcal{H}\big[\frac{1}{M}\sum_{m=1}^{M}\text{Prob}(\mathbf{y}_i|\mathbf{r};\boldsymbol{\theta}^{(m)})\big] - \frac{1}{M}\sum_{m=1}^{M}\mathcal{H}\big[(\mathbf{y}_i|\mathbf{r};\boldsymbol{\theta}^{(m)})\big], \quad \boldsymbol{\theta}^{(m)} \sim q(\boldsymbol{\theta})
\end{aligned}
$$

## C.2 JOINT PROBABILITY

At the test stage, we infer the joint probability by:

$$
\begin{aligned}
p(\mathbf{y}|\mathbf{r};\mathcal{G}) &= \int\int\text{Prob}(\mathbf{y}|\mathbf{p})\text{Prob}(\mathbf{p}|\mathbf{r};\boldsymbol{\theta})\text{Prob}(\boldsymbol{\theta}|\mathcal{G})d\mathbf{p}d\boldsymbol{\theta} \\
&\approx \int\int\text{Prob}(\mathbf{y}|\mathbf{p})\text{Prob}(\mathbf{p}|\mathbf{r};\boldsymbol{\theta})q(\boldsymbol{\theta})d\mathbf{p}d\theta \\
&\approx \frac{1}{M}\sum_{m=1}^{M}\int\text{Prob}(\mathbf{y}|\mathbf{p})\text{Prob}(\mathbf{p}|\mathbf{r};\boldsymbol{\theta}^{(m)})d\mathbf{p}, \quad \boldsymbol{\theta}^{(m)} \sim q(\boldsymbol{\theta}) \\
&\approx \frac{1}{M}\sum_{m=1}^{M}\int\sum_{i=1}^{N}\text{Prob}(\mathbf{y}_i|\mathbf{p}_i)\text{Prob}(\mathbf{p}_i|\mathbf{r};\boldsymbol{\theta}^{(m)})d\mathbf{p}_i, \quad \boldsymbol{\theta}^{(m)} \sim q(\boldsymbol{\theta}) \\
&\approx \frac{1}{M}\sum_{m=1}^{M}\sum_{i=1}^{N}\int\text{Prob}(\mathbf{y}_i|\mathbf{p}_i)\text{Prob}(\mathbf{p}_i|\mathbf{r};\boldsymbol{\theta}^{(m)})d\mathbf{p}_i, \quad \boldsymbol{\theta}^{(m)} \sim q(\boldsymbol{\theta}) \\
&\approx \frac{1}{M}\sum_{m=1}^{M}\prod_{i=1}^{N}\int\text{Prob}(\mathbf{y}_i|\mathbf{p}_i)\text{Dir}(\mathbf{p}_i|\boldsymbol{\alpha}_i^{(m)})d\mathbf{p}_i, \quad \boldsymbol{\alpha}^{(m)} = f(\mathbf{r},\boldsymbol{\theta}^{(m)}), q \quad \boldsymbol{\theta}^{(m)} \sim q(\boldsymbol{\theta})
\end{aligned}
$$

where the posterior over class label $p$ will be given by the mean of the Dirichlet:

$$\text{Prob}(y_i = p|\boldsymbol{\theta}^{(m)}) = \int \text{Prob}(y_i = p|\mathbf{p}_i)\text{Prob}(\mathbf{p}_i|\mathbf{r};\boldsymbol{\theta}^{(m)})d\mathbf{p}_i = \frac{\alpha_{ip}^{(m)}}{\sum_{k=1}^{K}\alpha_{ik}^{(m)}}$$

The probabilistic form for a specific node $i$ by using marginal probability,

$$
\begin{aligned}
\text{Prob}(\mathbf{y}_i|\mathbf{r};\mathcal{G}) &= \sum_{y\backslash y_i} \text{Prob}(\mathbf{y}|\mathbf{r};\mathcal{G}) \\
&= \sum_{y\backslash y_i} \int\int \prod_{j=1}^{N} \text{Prob}(\mathbf{y}_j|\mathbf{p}_j)\text{Prob}(\mathbf{p}_j|\mathbf{r};\boldsymbol{\theta})\text{Prob}(\boldsymbol{\theta}|\mathcal{G})d\mathbf{p}d\boldsymbol{\theta} \\
&\approx \sum_{y\backslash y_i} \int\int \prod_{j=1}^{N} \text{Prob}(\mathbf{y}_j|\mathbf{p}_j)\text{Prob}(\mathbf{p}_j|\mathbf{r};\boldsymbol{\theta})q(\boldsymbol{\theta})d\mathbf{p}d\boldsymbol{\theta} \\
&\approx \sum_{m=1}^{M}\sum_{y\backslash y_i} \int \prod_{j=1}^{N} \text{Prob}(\mathbf{y}_j|\mathbf{p}_j)\text{Prob}(\mathbf{p}_j|\mathbf{r};\boldsymbol{\theta}^{(m)})d\mathbf{p}, \quad \boldsymbol{\theta}^{(m)}\sim q(\boldsymbol{\theta}) \\
&\approx \sum_{m=1}^{M}\Big[\sum_{y\backslash y_i} \int \prod_{j=1}^{N} \text{Prob}(\mathbf{y}_j|\mathbf{p}_j)\text{Prob}(\mathbf{p}_j|\mathbf{r};\boldsymbol{\theta}^{(m)})d\mathbf{p}_j\Big], \quad \boldsymbol{\theta}^{(m)}\sim q(\boldsymbol{\theta}) \\
&\approx \sum_{m=1}^{M}\Big[\sum_{y\backslash y_i} \prod_{j=1,j\neq i}^{N} \text{Prob}(\mathbf{y}_j|\mathbf{r}_j;\boldsymbol{\theta}^{(m)})\Big]\text{Prob}(\mathbf{y}_i|\mathbf{r};\boldsymbol{\theta}^{(m)}), \quad \boldsymbol{\theta}^{(m)}\sim q(\boldsymbol{\theta}) \\
&\approx \sum_{m=1}^{M} \int \text{Prob}(\mathbf{y}_i|\mathbf{p}_i)\text{Prob}(\mathbf{p}_i|\mathbf{r};\boldsymbol{\theta}^{(m)})d\mathbf{p}_i, \quad \boldsymbol{\theta}^{(m)}\sim q(\boldsymbol{\theta})
\end{aligned}
$$

specifically for probability of label $p$,

$$\text{Prob}(y_i = p|\mathbf{r};\mathcal{G}) \approx \frac{1}{M}\sum_{m=1}^{M} \frac{\alpha_{ip}^{(m)}}{\sum_{k=1}^{K}\alpha_{ik}^{(m)}}, \quad \boldsymbol{\alpha}^{(m)} = f(\mathbf{r},\boldsymbol{\theta}^{(m)}), \quad \boldsymbol{\theta}^{(m)}\sim q(\boldsymbol{\theta})$$

## C.3 KL-DIVERGENCE

KL-divergence between $\text{Prob}(\mathbf{y}|\mathbf{r};\boldsymbol{\beta})$ and $\text{Prob}(\mathbf{y}|\mathbf{r};\boldsymbol{\gamma},\mathcal{G})$:

$$
\begin{aligned}
\text{KL}[\text{Prob}(\mathbf{y}|\mathbf{r};\mathcal{G})||\text{Prob}(\mathbf{y}|\mathbf{r};\beta))] &= \mathbb{E}_{\text{Prob}(\mathbf{y}|\mathbf{r};\mathcal{G})}\Big[\log\frac{\text{Prob}(\mathbf{y}|\mathbf{r};\mathcal{G})}{\text{Prob}(\mathbf{y}|\mathbf{r};\beta)}\Big] \\
&\approx \mathbb{E}_{\text{Prob}(\mathbf{y}|\mathbf{r};\mathcal{G})}\Big[\log\frac{\prod_{i=1}^{N}\text{Prob}(\mathbf{y}_i|\mathbf{r};\mathcal{G})}{\prod_{i=1}^{N}\text{Prob}(\mathbf{y}_i|\mathbf{r};\beta)}\Big] \\
&\approx \mathbb{E}_{\text{Prob}(\mathbf{y}|\mathbf{r};\mathcal{G})}\Big[\sum_{i=1}^{N}\log\frac{\text{Prob}(\mathbf{y}_i|\mathbf{r};\mathcal{G})}{\text{Prob}(\mathbf{y}_i|\mathbf{r};\beta)}\Big] \\
&\approx \sum_{i=1}^{N}\mathbb{E}_{\text{Prob}(\mathbf{y}|\mathbf{r};\mathcal{G})}\Big[\log\frac{\text{Prob}(\mathbf{y}_i|\mathbf{r};\mathcal{G})}{\text{Prob}(\mathbf{y}_i|\mathbf{r};\beta)}\Big] \\
&\approx \sum_{i=1}^{N}\sum_{j=1}^{K}\text{Prob}(y_i = j|\mathbf{r};\mathcal{G})\Big(\log\frac{\text{Prob}(y_i = j|\mathbf{r};\mathcal{G})}{\text{Prob}(y_i = j|\mathbf{r};\beta)}\Big)
\end{aligned}
$$

The KL divergence between two Dirichlet distributions $\text{Dir}(\alpha)$ and $\text{Dir}(\hat{\alpha})$ can be obtained in closed form as follows:

$$\text{KL}[\text{Dir}(\alpha)||\text{Dir}(\hat{\alpha})] = \ln\Gamma(S) - \ln\Gamma(\hat{S}) + \sum_{c=1}^{K}\big(\ln\Gamma(\hat{\alpha}_c) - \ln\Gamma(\alpha_c)\big) + \sum_{c=1}^{K}(\alpha_c - \hat{\alpha}_c)(\psi(\alpha_c) - \psi(S))$$

where $S = \sum_{c=1}^{K}\alpha_c$ and $\hat{S} = \sum_{c=1}^{K}\hat{\alpha}_c$

