# OpenReview forum: "Uncertainty-Aware Prediction for Graph Neural Networks"
_ICLR.cc/2020/Conference — Reject_

### Official Review · AnonReviewer2 · 2019-10-23
**Official Blind Review #2**

**Rating:** 3

**Review:**

The authors proposed a Bayesian graph neural network framework for node classification. The proposed models outperformed the baselines in six node classification tasks. The main contribution is to evaluate various uncertainty measures for the uncertainty analysis of Bayesian graph neural networks. The authors show that vacuity and aleatoric measure are important to detect out-of-distribution and the dissonance uncertainty plays a key role for improving performance.

** Introduction/Conclusion/Contribution
- Applying MC dropout to graph neural networks is a bit old idea [1-5], it cannot be considered as the “contribution” of this study.
- The authors should emphasize more on what’s the advancements over existing studies.

** Methodology
- My understanding is the authors proposed to use multiple uncertainties (vacuity, dissonance, aleatoric, epstemic). Vacuity and dissonance measures can also be implemented with other Bayesian graph neural networks.
- Ablation study is need to demonstrate the usefulness of each component in the objective function (equation 11).
- Can “BGAT-T” be considered as the proposed method? It doesn't use the proposed GNN framework but just an extension of original GAT with MC dropout and knowledge distillation.

** Experiments
- Please provide the details of hyperparameter settings, e.g. optimizer, batch size, learning rates, …
- The authors should perform experiments with varying numbers of labels per category (e.g. 5, 10, 20), because different methods show different behaviors under label scarcity.
- There are a number of recently published papers that address node classification based on Bayesian graph neural networks, e.g., see references. They should be used as baselines if available.
- I think the authors should evaluate BGAT-T with Co.Physics, Ama.Computer, and Ama.Photo datasets, because this method was overall superior to other methods on the first three datasets.
- Experiments are insufficient in the uncertainty analysis (section 5). The authors can evaluate the performance of BGAT as well as other Bayesian graph neural networks in terms of uncertainty quantification performance.
- Also, in order to evaluate uncertainty quantification performance, I would suggest to look at the trade-off between classification accuracy and classification rejection based on the uncertainty, like accuracy-rejection curve in [6]

** misc
- Check the following sentence of subsection 3.6. “our key contribution is that the proposed Bayesian GNN model is capable of estimating various uncertainty types to predict existing GNNs.”

References
[1] Ryu, S., Kwon, Y., & Kim, W. Y. (2019). Uncertainty quantification of molecular property prediction with Bayesian neural networks. arXiv preprint arXiv:1903.08375.
[2] Zhang, Y., & Lee, A. A. (2019). Bayesian semi-supervised learning for uncertainty-calibrated prediction of molecular properties and active learning. arXiv preprint arXiv:1902.00925.
[3] Pal, S., Regol, F. L. O. R. E. N. C. E., & Coates, M. A. R. K. (2019). Bayesian graph convolutional neural networks using non-parametric graph learning. In Representation Learning on Graphs and Manifolds Workshop, Int. Conf. Learning Representations.
[4] Akita, H., Nakago, K., Komatsu, T., Sugawara, Y., Maeda, S. I., Baba, Y., & Kashima, H. (2018, December). Bayesgrad: Explaining predictions of graph convolutional networks. In International Conference on Neural Information Processing (pp. 81-92). Springer, Cham.
[5] Zhang, Y., Pal, S., Coates, M., & Ustebay, D. (2019, July). Bayesian graph convolutional neural networks for semi-supervised classification. In Proceedings of the AAAI Conference on Artificial Intelligence (Vol. 33, pp. 5829-5836).
[6] Nadeem, M. S. A., Zucker, J. D., & Hanczar, B. (2009, March). Accuracy-rejection curves (ARCs) for comparing classification methods with a reject option. In Machine Learning in Systems Biology (pp. 65-81).


**Experience Assessment:**

I have read many papers in this area.

**Review Assessment: Checking Correctness Of Derivations And Theory:**

I assessed the sensibility of the derivations and theory.

**Review Assessment: Checking Correctness Of Experiments:**

I carefully checked the experiments.

**Review Assessment: Thoroughness In Paper Reading:**

I read the paper at least twice and used my best judgement in assessing the paper.

---

> ### Author Response · Authors · 2019-11-15
> **Response to Reviewer #2 [Part 3]**
>
> Q8.  I think the authors should evaluate BGAT-T with Co.Physics, Ama.Computer, and Ama.Photo datasets, because this method was overall superior to other methods on the first three datasets.
>
> We agree with you. However, unfortunately, due to memory limitation (only 1 Nvidia Titan X GPU), we could not run the BGAT-T on Co.Physics, Ama.Computer, and Ama.Photo, which are very dense datasets.
>
> Q9.  Experiments are insufficient in the uncertainty analysis (section 5). The authors can evaluate the performance of BGAT as well as other Bayesian graph neural networks in terms of uncertainty quantification performance.
>
> Since the purpose of this uncertainty analysis is to investigate the impact of different types of uncertainty estimates, we should assume that all baseline methods need to use the same GNN model (e.g., GCN in our work).  That is why it is not feasible to consider all other Bayesian GNNs in terms of uncertainty quantification performance.  For GAT related model, we will add its uncertainty experimental results in the appendix.
>
> Q10.  Also, in order to evaluate uncertainty quantification performance, I would suggest to look at the trade-off between classification accuracy and classification rejection based on the uncertainty, like accuracy-rejection curve in [6]
>
> Our metric "Precision-Recall curve" (also used in [8]) in Figure 2 (Section 5.1) is exactly the same as accuracy-rejection curve [6].  Recall = 1.0 means accepting all (no reject) testing samples while recall = 0 means rejecting all testing samples (accuracy or precision equal to 1), while Figure 2 only shows PR curves with recall range [0.05-1]. As shown in Figure 2, using dissonance performed the best among all in PR curves.
>
> References
> [6] Nadeem, M. S. A., Zucker, J. D., & Hanczar, B. (2009, March). Accuracy-rejection curves (ARCs) for comparing classification methods with a reject option. In Machine Learning in Systems Biology (pp. 65-81).
> [8] Kendall, Alex, and Yarin Gal. "What uncertainties do we need in bayesian deep learning for computer vision?." Advances in neural information processing systems (NIPS). 2017.

---

> ### Author Response · Authors · 2019-11-15
> **Response to Reviewer #2 [Part 2]**
>
> Q4.  Can “BGAT-T” be considered as the proposed method? It doesn't use the proposed GNN framework but just an extension of original GAT with MC dropout and knowledge distillation.
> Yes, `BGAT-T' is our proposed method based on the proposed multidimensional uncertainty estimates as a variant of the original GAT model (similar as S-BGCN-T-K) and re-named as "S-BGAT-T-K" in our revised paper.  Our proposed uncertainty framework is highly applicable across all GNNs.  The key novelty of our proposed S-BAGT-T is taking a Bayesian framework to estimate probabilistic uncertainty (i.e., aleatoric and epistemic uncertainty) and outputs Dirichlet distribution, instead of using softmax, in order to estimate evidential uncertainty (i.e., vacuity and dissonance) as well.  We detailed the design process of BAGT-T (S-BGAT-T-K) in Fig. 1.
>
> Q5.  Please provide the details of hyperparameter settings, e.g. optimizer, batch size, learning rates, …
>
> We showed detailed hyperparameter settings in Appendix A, along with Table 6 and Table 7.
>
> Q6.  The authors should perform experiments with varying numbers of labels per category (e.g. 5, 10, 20), because different methods show different behaviors under label scarcity.
>
> The key contribution of this work is not to show the improvement of semi-supervised node classification accuracy as many other works do.  The focus of our work is to estimate different types of uncertainty and how they can be meaningfully used for decision making.  That is, we use this node classification accuracy as a proxy metric in order to show an example on how the estimates of multidimensional uncertainty can possibly improve decision performance and the interplay between the different types of uncertainty and their impact on the decision performance (i.e., classification accuracy).  To this end, we showed the performance comparison in the classification accuracy in Table 3.
>
> Q7.  There are a number of recently published papers that address node classification based on Bayesian graph neural networks, e.g., see references. They should be used as baselines if available.
>
> Bayesian framework is not our contribution, but is used to estimate probabilistic uncertainty such as aleatoric uncertainty and epistemic uncertainty.  Since our work focuses on measuring different types of uncertainty, in which we call it `uncertainty decomposition,' we didn't compare all [1-5] with our schemes. However, since [7] studied the uncertainty decomposition by proposing `distributional uncertainty,' we compared it with our models as a baseline for OOD detection task.  Since we also showed the effect of different uncertainty types estimated on classification accuracy, we considered [3, 5] for testing the accuracy of semi-supervised node classification in Table 2 as the names of "Bayesian GCN" and "BGCNN", respectively.  Our results in Table 2 showed that our proposed S-BGCN-T outperformed BGCN and BGCNN [3, 5] in the Cora and Pubmed datasets while slightly less performed on the Citeseer dataset.  Our baseline Bayesian GCN model, named GCN-Drop (using MC-Dropout) is the same as [1], which is compared with our proposed model.  More importantly, [2-5] only considered overall uncertainty (e.g., entropy or variance) and [1] only considered aleatoric and epistemic uncertainty.  On the other hand, our uncertainty experiments (see Section 5) already considered overall uncertainty, aleatoric and epistemic uncertainty as baselines, such as GCN-Drop, which is also used in [1].
>
>
> References
> [1] Ryu, S., Kwon, Y., & Kim, W. Y. (2019). Uncertainty quantification of molecular property prediction with Bayesian neural networks. arXiv preprint arXiv:1903.08375.
> [2] Zhang, Y., & Lee, A. A. (2019). Bayesian semi-supervised learning for uncertainty-calibrated prediction of molecular properties and active learning. arXiv preprint arXiv:1902.00925.
> [3] Pal, S., Regol, F. L. O. R. E. N. C. E., & Coates, M. A. R. K. (2019). Bayesian graph convolutional neural networks using non-parametric graph learning. In Representation Learning on Graphs and Manifolds Workshop, Int. Conf. Learning Representations.
> [4] Akita, H., Nakago, K., Komatsu, T., Sugawara, Y., Maeda, S. I., Baba, Y., & Kashima, H. (2018, December). Bayesgrad: Explaining predictions of graph convolutional networks. In International Conference on Neural Information Processing (pp. 81-92). Springer, Cham.
> [5] Zhang, Y., Pal, S., Coates, M., & Ustebay, D. (2019, July). Bayesian graph convolutional neural networks for semi-supervised classification. In Proceedings of the AAAI Conference on Artificial Intelligence (Vol. 33, pp. 5829-5836).
> [7] Malinin, Andrey, and Mark Gales. "Predictive uncertainty estimation via prior networks." Advances in Neural Information Processing Systems (NIPS). 2018.

---

> ### Author Response · Authors · 2019-11-15
> **Response to Reviewer #2 [Part 1]**
>
> Q1.  Applying MC dropout to graph neural networks is a bit old idea [1-5], it cannot be considered as the “contribution” of this study.  The authors should emphasize more on what’s the advancements over existing studies.
>
> We agree with you. We better emphasized the key contribution of our work in Section 1 by adding the following (a second bullet point of the key contributions): "We propose a Graph-based Kernel Dirichlet distribution Estimation (GKDE) method to reduce error in predicting Dirichlet distribution. We designed an iterative knowledge distillation algorithm that treats a deterministic GNN as a Teacher Network while considering our proposed Subjective Bayesian GNN model (a realization of our proposed framework for a specific GNN) as a distilled network.  We conducted an in-depth comparative performance analysis of our proposed model and existing counterparts which only considered probabilistic uncertainty [1] or total uncertainty (entropy or variance) [2-5].  The experimental results prove that our proposed model dealing with both probabilistic and evidential uncertainty can significantly outperform the existing counterparts in both node classification and OOD detection as shown in Tables 3 and 4."
>
> Q2.  My understanding is the authors proposed to use multiple uncertainties (vacuity, dissonance, aleatoric, epistemic). Vacuity and dissonance measures can also be implemented with other Bayesian graph neural networks.
>
> This is not true at all.  Let us discuss the key contribution, in relation with what Bayesian GNNs can (cannot) do and what our proposed framework can do.  Bayesian GNNs can only estimate aleatoric uncertainty and epistemic uncertainty, which has been addressed in the existing predictive uncertainty research.  These two uncertainty types are mainly related to uncertainty in data and model/system which is derived from statistical process and its nature in randomness. So we call them "probabilistic uncertainty".  In our proposed method, we measure the so called "evidential uncertainty" which is studied in terms of vacuity (uncertainty due to a lack of evidence) and dissonance (uncertainty due to conflicting evidence) in a belief/evidence theory research community.  These evidential uncertainty types cannot be estimated from Bayesian GNNs; on the other hand, our proposed method outputs the Dirichlet distribution, instead of using softmax, which allows us to estimate vacuity and dissonance.  In addition, our method also takes a Bayesian framework in order to estimate the probabilistic uncertainty types, aleatoric and epistemic uncertainty.  Therefore, our proposed method offers the capability to estimate all four uncertainty types and use them to improve classification prediction accuracy. On the other hand, the Bayesian GNN can only estimate aleatoric and epistemic uncertainty types.
>
> Q3.  Ablation study is need to demonstrate the usefulness of each component in the objective function (equation 11).
>
> We conducted an additional experiment to ensure the benefit of the teacher network.  We anticipate that the graph kernel prior will improve the estimation accuracy of Dirichlet distribution. However, due to the space constraint, we didn't show the classification results without using the graph kernel prior.  In the revised version, we added a detailed ablation study in the revised paper in order to clearly demonstrate the contribution of the key technical components, including a teacher Network, Graph kernel Dirichlet Estimation (GKDE) and subjective Bayesian framework.  The key findings obtained from this experiment are: (1) The teacher Network can further improve node classification accuracy (i.e., 0.2%  - 1.5% increase, as shown in Table 2); and (2) GKDE (graph kernel prior) using the uncertainty estimates can enhance OOD detection (i.e., 4% - 7% increase, as shown in Table 9).
>
>
> References
> [1] Ryu, S., Kwon, Y., & Kim, W. Y. (2019). Uncertainty quantification of molecular property prediction with Bayesian neural networks. arXiv preprint arXiv:1903.08375.
> [2] Zhang, Y., & Lee, A. A. (2019). Bayesian semi-supervised learning for uncertainty-calibrated prediction of molecular properties and active learning. arXiv preprint arXiv:1902.00925.
> [3] Pal, S., Regol, F. L. O. R. E. N. C. E., & Coates, M. A. R. K. (2019). Bayesian graph convolutional neural networks using non-parametric graph learning. In Representation Learning on Graphs and Manifolds Workshop, Int. Conf. Learning Representations.
> [4] Akita, H., Nakago, K., Komatsu, T., Sugawara, Y., Maeda, S. I., Baba, Y., & Kashima, H. (2018, December). Bayesgrad: Explaining predictions of graph convolutional networks. In International Conference on Neural Information Processing (pp. 81-92). Springer, Cham.
> [5] Zhang, Y., Pal, S., Coates, M., & Ustebay, D. (2019, July). Bayesian graph convolutional neural networks for semi-supervised classification. In Proceedings of the AAAI Conference on Artificial Intelligence (Vol. 33, pp. 5829-5836).

---

### Official Review · AnonReviewer1 · 2019-10-24
**Official Blind Review #1**

**Rating:** 3

**Review:**

This paper proposes to model various uncertainty measures in Graph Convolutional Networks (GCN) by Bayesian MC Dropout. Compared to existing Bayesian GCN methods, this work stands out in two aspects: 1) in terms of prediction, it considers multiple uncertainty measures including aleatoric, epistemic, vacuity and dissonance (see paper for definitions); 2) in terms of generative modeling, the GCN first predicts the parameters of a Dirichlet distribution, and then the class probabilities are sampled from the Dirichlet. Training/inference roughly follows MC Dropout, with two additional priors/teachers: 1) the prediction task is guided by a deterministic teacher network (via KL(model || teacher)), and 2) the Dirichlet parameters are guided by a kernel-based prior (via KL(model || prior)). Experiments on six datasets showed superior performance in terms of the end prediction task, as well as better uncertainty modeling in terms of out-of-distribution detection.

Pros:
1. This model considers uncertainties in multiple dimensions.
2. Better predictive performance and OOD detection ability on 6 real datasets.

Cons:
1. Adding an additional layer of the Dirichlet is not well motivated.
2. Needs ablation studies on modeling choices, e.g., how much does the graph kernel prior help.
3. In table 2, needs to compare with traditional Bayesian GCN (such as Zhang et al 2018). Besides, is GCN-Drop in table 4 Zhang et al 2018?
4. In table 2, seems that knowledge distillation helps, since GCN gets similar performance to BGCN, but BGCN-T outperforms. A natural baseline is GCN w/ knowledge distillation.
5. In table 4, the baseline GCN-Drop gets better uncertainty estimates than the proposed approach in terms of aleatoric, epistemic, entropy which can be evaluated for GCN-Drop. I wonder if it is possible to develop a measure for vacuity and dissonance for GCN-Drop as well. But anyway this table contradicts the motivation for adding an additional layer of the Dirichlet.

Minor details:
1. Is teacher jointly trained with the model or is it pretrained? And what's teacher's network architecture? Is it much larger? I cannot understand "choose two graph convolutional layers in which the first layer is 16hidden units for GCN and 64 hidden units for GAT, and removed a softmax layer".

Overall, this is a very technical work, but the modeling choices need to be better justified/motivated compared to existing works on GCN with MC Dropout. I am inclined to reject this paper.


-------updates after reading rebuttal----
Thanks for the response! I have no further questions.

**Experience Assessment:**

I have read many papers in this area.

**Review Assessment: Checking Correctness Of Derivations And Theory:**

I carefully checked the derivations and theory.

**Review Assessment: Checking Correctness Of Experiments:**

I carefully checked the experiments.

**Review Assessment: Thoroughness In Paper Reading:**

I read the paper thoroughly.

---

> ### Author Response · Authors · 2019-11-15
> **Response to Reviewer #1 [Part 2]**
>
> Q5.  In table 4, the baseline GCN-Drop gets better uncertainty estimates than the proposed approach in terms of aleatoric, epistemic, entropy which can be evaluated for GCN-Drop. I wonder if it is possible to develop a measure for vacuity and dissonance for GCN-Drop as well. But anyway this table contradicts the motivation for adding an additional layer of the Dirichlet.
>
> First of all, it is not true that GCN-Drop outperforms our proposed method in terms of the three uncertainty estimates.  In Table 4, we can clearly observe that our proposed method BGCB-T (renamed as S-BGCN-T-K in the revised paper) outperforms GCN-Drop in terms of the estimates of aleatoric and entropy.  However, our proposed method is slightly less performing than GCN-Drop in the estimates of epistemic uncertainty.  This is because the epistemic uncertainty is not an appropriate uncertainty metric in OOD detection.  This is observed clearly as the epistemic estimates show much lower performance (e.g., ~50-70 in AUROC and 40-60 in AUPR) than other uncertainty estimates.
>
> In GNN models, no uncertainty metrics have been developed.  For better, fair comparison, we devised the baseline models using GNN models. So the baseline we created, named GCN-Drop, is an adapted version of a GNN model to Bayesian framework and used MC-Drop to infer the model, which allowed us to derive aleatoric and epistemic uncertainty estimates.  If we add the estimates of vacuity and dissonance in GCN-Drop, this will be our proposed BGCN (renamed as S-BGCN-T-K in the revised paper), which outputs Dirichlet distribution, instead of class probabilities.  Again, the key motivation of our method using additional layers of the Dirichlet distribution is to estimate the multidimensional uncertainty whose dimensions include both probabilistic uncertainty (i.e., aleatoric uncertainty and epistemic uncertainty) and evidential probability (i.e., vacuity and dissonance).
>
>
> Q6.  Is teacher jointly trained with the model or is it pretrained? And what's teacher's network architecture? Is it much larger? I cannot understand "choose two graph convolutional layers in which the first layer is 16hidden units for GCN and 64 hidden units for GAT, and removed a softmax layer".
>
> A Teacher Network (GNN) can be jointly trained or pre-trained with the student network (S-BGNN), where the pre-trained model achieves better performance in practice.  The Teacher Network is the original GNN (e.g., GCN, GAT).  S-BGCN (our proposed model) is designed based on original GCN with two graph convolution layers with the first layer being 16 hidden units for Cora, Citeseer, Pubmed datasets.  S-BGAT (our proposed model) is designed based on original GAT with two graph convolution layer with the first layer being 64 hidden units. We made this clear in the revised paper on table 6 and table 7.

---

> ### Author Response · Authors · 2019-11-15
> **Response to Reviewer #1 [Part 1]**
>
> Q1.  Adding an additional layer of the Dirichlet is not well motivated.
>
> The key theme of this work is to consider uncertainty in the node classification as a decision making problem where the uncertainty may be derived from different root causes.  By bridging the uncertainty research in ML/DL and evidence/belief theory (i.e., Subjective Logic), we aimed to estimate four different types of uncertainty embracing probabilistic uncertainty (i.e., aleatoric uncertainty and epistemic uncertainty) in ML/DL and evidential uncertainty (i.e., vacuity and dissonance) in Subjective Logic and investigate its usefulness in improving the effectiveness of decision making.  Specifically, we proposed a Subjective Bayesian Graph Neural Networks (S-BGNN), which directly outputs Dirichlet distribution parameter $\alpha$, instead of a softmax probability, in which Bayesian framework allows us to estimate the probabilistic uncertainty while the Dirichlet distribution-based multinomial subjective opinions in SL enabled us to estimate the evidential uncertainty.  We made this clear in the revised paper (Section 1) in blue.
>
> Q2.  Needs ablation studies on modeling choices, e.g., how much does the graph kernel prior help.
>
> We conducted an additional experiment to ensure the benefit of the teacher network.  We anticipate that the graph kernel prior will improve the estimation accuracy of Dirichlet distribution. However, due to the space constraint, we didn't show the classification results without using the graph kernel prior.  In the revised version, we added a detailed ablation study in the revised paper in order to clearly demonstrate the contribution of the key technical components, including a teacher Network, Graph kernel Dirichlet Estimation (GKDE) and subjective Bayesian framework.  The key findings obtained from this experiment are: (1) The teacher Network can further improve node classification accuracy (i.e., 0.2%  - 1.5% increase, as shown in Table 2); and (2) GKDE (graph kernel prior) using the uncertainty estimates can enhance OOD detection (i.e., 4% - 7% increase, as shown in Table 9).
>
>
> Q3.  In table 2, needs to compare with traditional Bayesian GCN (such as Zhang et al 2018). Besides, is GCN-Drop in table 4 Zhang et al 2018?
>
> The key purpose of our work is to show how much the estimation of multidimensional uncertainty-aware classification process can help decision making performance.  In our submitted paper, the reason of not showing the classification prediction accuracy of Zhang et al. (2018)'s work was because we focused on the estimation of multidimensional uncertainty. However, if the uncertainty estimates does not give positive impact on prediction accuracy, it may not be meaningful.  Hence, reflecting your comments, we added the prediction accuracy of Zhang et al. (2018)'s work in the revised paper on table 2 (named "Bayesian GCN").  Using the same popular datasets, including Cora, Citeseer, Pubmed, with a fixed setting. We observed the following results, which are also added in Table 2 of the revised paper. As you observe in the below, our method showed outperformance over Zhang et al. (2018) in Cora and Pubmed while comparably performed with Citeseer.  Notice that GCN-Drop in Table 4 is the same as [1], not Zhang et al. (2018)'s work.
> ---------------------------------------------------------------------------------
>                                       Cora              Citeseer             Pubmed
> Zhang et al 2018        81.2±0.8        72.2±0.6            76.6±0.7
> Ours(S-BGCN-T)         82.0±0.7        71.2±0.8           79.3±0.4
> --------------------------------------------------------------------------------
>
> Q4.  In table 2, seems that knowledge distillation helps, since GCN gets similar performance to BGCN, but BGCN-T outperforms. A natural baseline is GCN w/ knowledge distillation.
>
> BGCN-T  (renamed as S-BGCN-T-K in the revised paper) uses a GCN (a discriminative model) as a Teacher Network for knowledge distillation. Based on our understanding, GCN cannot be used with knowledge distillation by itself. Hence, the natural baseline model is a GCN, as we considered in our work.
>
>
> References
> [1] Ryu, S., Kwon, Y., & Kim, W. Y. (2019). Uncertainty quantification of molecular property prediction with Bayesian neural networks. arXiv preprint arXiv:1903.08375.

---

### Official Review · AnonReviewer3 · 2019-11-06
**Official Blind Review #3**

**Rating:** 3

**Review:**

This paper studies the effect of various uncertainties in bayesian GNNs. They study various uncertainty models such as leatoric, epistemic, vacuity and dissonance.  They study multiple uncertainty types in both
deep learning (DL) and belief/evidence theory domains. They treat the predictions
of a Bayesian GNN (BGNN) as nodes’ multinomial subjective opinions in a graph
based on Dirichlet distributions where each belief mass is a belief probability of
each class. By collecting evidence from the given labels of training nodes, the
BGNN model is designed for accurately predicting probabilities of each class and
detecting out-of-distribution. They show that their proposed Bayesian GNN outperforms state-of-the-art counterparts in terms of the accuracy of node classification prediction and out-of-distribution detection based on six real
network datasets.

The main issue with this paper is the motivation of the framework. While the technical details seem correct, the writing is unclear in many places in the manuscript. From an experimental perspective, ablation experiments need to be added. Also, the main motivation of this work is that it is modeling the uncertainty in model predictions. However, I do not see this verified anywhere experimentally.

**Experience Assessment:**

I have read many papers in this area.

**Review Assessment: Checking Correctness Of Derivations And Theory:**

I assessed the sensibility of the derivations and theory.

**Review Assessment: Checking Correctness Of Experiments:**

I assessed the sensibility of the experiments.

**Review Assessment: Thoroughness In Paper Reading:**

I made a quick assessment of this paper.

---

> ### Author Response · Authors · 2019-11-15
> **Response to Reviewer #3**
>
> Q 1. The main issue with this paper is the motivation of the framework. While the technical details seem correct, the writing is unclear in many places in the manuscript.
>
> We made clear the key contributions of our work as follows (also elaborated in the revised version in blue):
>
> 1.   Developed a Subjective Bayesian framework to estimate multiple dimensions of uncertainty in GNNs, embracing both probabilistic uncertainty (i.e., aleatoric and epistemic uncertainty) and evidential uncertainty (i.e., vacuity and dissonance), which has not been explored in the existing approaches.
> 2.   Proposed Graph-based kernel Dirichlet Estimation (GKDE) to help improve the uncertainty estimation based on Dirichlet distribution.
> 3.   Proposed a Teacher network to help improve the expected class probability estimation.
>
> In addition, we also added the following phrases in order to clarify the motivation of this research (see pp. 1-2 in blue): Our aim is to: (1) characterize inherent dimensions of uncertainty in data, derived from its different root causes; (2) develop a multidimensional uncertainty-aware framework, named “Subjective Bayesian Graph Neural Network” (S-BGNN) that can estimate the multidimensional uncertainty in graph neural networks (GNNs); and (3) demonstrate the outperformance of the proposed framework in classification prediction accuracy when multidimensional uncertainty is considered.  The existing Bayesian framework in deep learning model (including GNNs) only considers “probabilistic uncertainty”, such as aleatoric and epistemic uncertainty.  It does not have the ability for representing and interpreting uncertainty in terms of its different manifestations caused by unreliable, incomplete, deceptive, and conflicting information.  In our work, we introduce two additional “evidential uncertainty” types such as vacuity and dissonance, which is used in Subjective Logic, which is one of evidence/belief models that explicitly deals with uncertainty.  Our proposed multidimensional uncertainty-aware framework (i.e., S-BGNN) provides the ways of estimating these four types of uncertainty embracing both probabilistic and evidential uncertainty types by having a GNN directly output the parameters of Dirichlet distribution which allow us to measure the four different types of uncertainty.
>
> Q 2. From an experimental perspective, ablation experiments need to be added.
>
> We conducted an additional experiment to ensure the benefit of the teacher network.  We anticipate that the graph kernel prior will improve the estimation accuracy of Dirichlet distribution. However, due to the space constraint, we didn't show the classification results without using the graph kernel prior.  In the revised version, we added a detailed ablation study in the revised paper in order to clearly demonstrate the contribution of the key technical components, including a teacher Network, Graph kernel Dirichlet Estimation (GKDE) and subjective Bayesian framework.  The key findings obtained from this experiment are: (1) The teacher Network can further improve node classification accuracy (i.e., 0.2%  - 1.5% increase, as shown in Table 2); and (2) GKDE (graph kernel prior) using the uncertainty estimates can enhance OOD detection (i.e., 4% - 7% increase, as shown in Table 9).
>
> Q 3. Also, the main motivation of this work is that it is modeling the uncertainty in model predictions. However, I do not see this verified anywhere experimentally.
>
> We conducted extensive experiments to verify the proposed multidimensional uncertainty-aware DL framework that predicts multidimensional uncertainty including both probabilistic and evidential uncertainty types (aleatoric, epistemic, vacuity, and dissonance) in Section 5 (`Uncertainty Experiment and Analysis'). We summarized the key findings of the extensive experiments in Table 3, Table 4, Figure 2, and Figure 3 in terms of: (1) node classification, using Precision-Recall curve [1] as metric (similar as `accuracy-rejection curves' in [4] ), in which it is expected to observe that a reasonable uncertainty estimate is high for misclassification and low for correct prediction; and (2) out-of-distribution (OOD) detection, using AUPR and AUROC metrics (also used in [2, 3]). In addition, we compared our proposed model with several strong baselines (e.g., entropy uncertainty, epistemic uncertainty and distributional uncertainty). In this experiment, we observed high uncertainty for out-of-distribution sample and low uncertainty for in-distribution sample, which is intuitively expected as reasonable uncertainty estimates.
>
>
> References
> [1] Kendall, Alex, and Yarin Gal. "What uncertainties do we need in bayesian deep learning for computer vision?." NIPS 2017.
> [2] Malinin, Andrey, and Mark Gales. "Predictive uncertainty estimation via prior networks." NIPS  2018.
> [3] Hendrycks, Dan, and Kevin Gimpel. "A baseline for detecting misclassified and out-of-distribution examples in neural networks." ICLR 2017.

---

### Author Response · Authors · 2019-11-15
**Response to all reviewers**

We want to first thank all the reviewers for taking time to review our paper and providing their insightful comments and suggestions. We have revised the paper by following the reviewers' suggestions and made the major changes in blue in the revised paper for the easy location of the new changes.  Here is the overall summary of the changes made in the paper reflecting the review comments received.

1.   We made clear the motivation of our proposed model.  Our aim is to: (1) characterize inherent dimensions of uncertainty in data, derived from its different root causes; (2) develop a multidimensional uncertainty-aware framework, named “Subjective Bayesian Graph Neural Network” (S-BGNN) that can estimate the multidimensional uncertainty in graph neural networks (GNNs); and (3) demonstrate the outperformance of the proposed framework in classification prediction accuracy when multidimensional uncertainty is considered.  The existing Bayesian framework in deep learning model (including GNNs) only considers `probabilistic uncertainty,' such as aleatoric and epistemic uncertainty.  It does not have the ability for representing and interpreting uncertainty in terms of its different manifestations caused by unreliable, incomplete, deceptive, and conflicting information.  In our work, we introduce two additional `evidential uncertainty' types such as vacuity and dissonance, which is used in Subjective Logic, which is one of evidence/belief models that explicitly deals with uncertainty.  Our proposed multidimensional uncertainty-aware framework (i.e., S-BGNN) provides the ways of estimating these four types of uncertainty embracing both probabilistic and evidential uncertainty types by having a GNN directly output the parameters of Dirichlet distribution which allow us to measure the four different types of uncertainty.

The four types of uncertainty under two different categories are considered in this work as follows:
   a) Probabilistic uncertainty: The below two uncertainty types are measured by taking a Bayesian framework, as a traditional DL framework.
           Aleatoric uncertainty: Uncertainty is caused by noise inherent in data.
           Epistemic uncertainty: Uncertainty is introduced by imperfect model parameters.

  b) Evidential uncertainty: The below two uncertainty types are estimated by using a belief model, called Subjective Logic.
            Vacuity: Uncertainty is generated because of a lack of evidence.
            Dissonance: Uncertainty exists due to conflict evidence.

2.    To clarify the differences between our proposed model and the existing baseline counterpart models, we re-named our proposed methods in the revised paper.

GNN: Original Graph Neural Network with cross-entropy loss, which can only estimate entropy uncertainty (i.e., GCN, GAT).

BGNN: Original Graph Neural Network (cross-entropy loss) with Bayesian framework, in which the weight parameters follow a distribution, that allows us to estimate aleatoric and epistemic uncertainty (i.e., GCN-Drop).

S-GNN (Our proposed model): Subjective Graph Neural Network (with square-loss) that can estimate two evidential uncertainty types (vacuity and dissonance) by outputting subjective opinions (following Dirichlet distribution), instead of softmax probability (i.e.,  S-GCN, S-GAT).

S-BGNN (Our proposed model): Subjective Graph Neural Network (with square-loss) with the Bayesian framework that can estimate all four uncertainty types, including aleatoric uncertainty, epistemic uncertainty, vacuity and dissonance (i.e., S-BGCN, S-BGAT).

S-BGNN-T (Our proposed model): S-BGNN with a Teacher network to improve the expected class probability estimation (i.e., S-BGCN-T, S-BGAT-T).

S-BGNN-T-K (Our proposed model): S-BGNN-T with Graph kernel Dirichlet Estimation (GKDE) to improve the uncertainty (Dirichlet distribution) estimation (i.e., S-BGCN-T-K, S-BGAT-T-K).

3.    We conducted an additional experiment to ensure the benefit of the teacher network.  We anticipate that the graph kernel prior will improve the estimation accuracy of Dirichlet distribution. However, due to the space constraint, we didn't show the classification results without using the graph kernel prior.  In the revised version, we added a detailed ablation study in the revised paper in order to clearly demonstrate the contribution of the key technical components, including a teacher Network, Graph kernel Dirichlet Estimation (GKDE) and subjective Bayesian framework.  The key findings obtained from this experiment are: (1) The teacher Network can further improve node classification accuracy (i.e., 0.2%  - 1.5% increase, as shown in Table 2); and (2) GKDE (graph kernel prior) using the uncertainty estimates can enhance OOD detection (i.e., 4% - 7% increase, as shown in Table 9).

---

### Decision · Program_Chairs · 2019-12-19

**Decision:**

Reject

**Comment:**

The authors propose a way to produce uncertainty measures in graph neural networks. However, the reviewers find that the methods proposed lack novelty and are incremental additions to prior work.